# Apical constriction requires patterned apical surface remodeling to synchronize cellular deformation

**Satoshi Yamashita[1]\*, Shuji Ishihara[2†], François Graner[3†]**

[1]Laboratory for Morphogenetic Signaling, RIKEN Center for Biosystems Dynamics Research, Kobe, Japan; [2]Graduate School of Arts and Sciences, The University of Tokyo, Tokyo, Japan; [3]Université Paris Cité, CNRS, MSC, Paris, France

## eLife Assessment

The results from this study, which investigates the mechanisms necessary for initiating tissue invagination using a cellular Potts modelling approach, suggest that apical constriction is not sufficient to drive the process by itself. The study highlights how choices inherent to modeling – such as permitting straight or curved cell edges – may affect the outcome of simulations and, consequently, their biophysical interpretation. Despite **incomplete** evidence supporting their major claims due to a rather coarse-grained exploration of the model, this work is **useful** for biophysicists investigating complex tissue deformation through computational frameworks.

**\*For correspondence:**
satoshiy83@gmail.com

[†]These authors contributed equally to this work

**Competing interest:** The authors declare that no competing interests exist.

**Abstract** Apical constriction is a basic mechanism for epithelial morphogenesis, making columnar cells into wedge shape and bending a flat cell sheet. It has long been thought that an apically localized myosin generates a contractile force and drives the cell deformation. However, when we tested the increased apical surface contractility in a cellular Potts model simulation, the constriction increased pressure inside the cell and pushed its lateral surface outward, making the cells adopt a drop shape instead of the expected wedge shape. To keep the lateral surface straight, we considered an alternative model in which the cell shape was determined by cell membrane elasticity and endocytosis, and the increased pressure is balanced among the cells. The cellular Potts model simulation succeeded in reproducing the apical constriction, and it also suggested that a too strong apical surface tension might prevent the tissue invagination.

## Introduction

The apical constriction is a general and fundamental process to bend or invaginate a flat epithelial tissue (*Martin and Goldstein, 2014*). In the flat epithelium, cells are in a columnar shape with apical-basal polarity. During the apical constriction, the columnar cells decrease their apical surface and adopt a wedge shape, inducing a small angle between its opposite lateral surfaces. This cellular deformation is synchronized in a cluster of cells, and the angle of the lateral surfaces is summed across the cells to a larger tissue scale curve. Studies in genetics and molecular biology had revealed an involvement of the motor protein non-muscle myosin II which crosslinks and contracts the actin filament network. The actomyosin network is formed beneath the cell apical surface (cortical actomyosin) and lining apical-lateral cell-cell interface (circumferential actomyosin), linked with adherens junctions and tight junctions so that it makes a supracellular continuous structure (*Martin et al., 2010*; *Takeichi, 2014*).

In *Drosophila melanogaster* mesoderm invagination, ventral cells express transcription factors Twist and Snail which then induce expression of numerous genes including regulators of the actin and

myosin, such as T48, Fog, Mist, RhoGEF2, Rho, and Rock (*Martin, 2020*). The cortical and the circumferential actomyosin network are formed in the cells, and the cells undergo the apical constriction, while down-regulation of those regulators impaired the localization and activation of the myosin, and delayed or prevented the apical constriction (*Dawes-Hoang et al., 2005*; *Xie et al., 2016*; *Kölsch et al., 2007*; *Mason et al., 2013*). Similar regulation of the myosin by Rho and Rock was found responsible for the apical constriction in other organs (*Brodu and Casanova, 2006*; *Chung et al., 2017*; *Guru et al., 2022*; *Manning et al., 2013*; *Röper, 2012*) and other insects (*Münster et al., 2019*; *Benton et al., 2019*; *Urbansky et al., 2016*).

In vertebrates, the cortical myosin was not observed but the circumferential myosin was enriched, recruited together with Rock by PDZ domain containing protein Shroom 3, which was also found responsible for the apical constriction (*Nishimura and Takeichi, 2008*; *Lang et al., 2014*; *Plageman et al., 2010*; *Lee et al., 2007*; *Chung et al., 2010*; *McGreevy et al., 2015*; *Haigo et al., 2003*; *Hildebrand, 2005*).

Since the interruption of those pathways reduced the myosin accumulation and prevented the apical constriction, it had been modeled that the apical myosin generated a contractile force on the cell apical surface and drove the cell and tissue deformation.

The myosin-based model was validated by simulations (*Rauzi et al., 2013*) with finite element model (*Conte et al., 2012*; *Perez-Mockus et al., 2017*) or vertex model (*Sherrard et al., 2010*; *Polyakov et al., 2014*; *Inoue et al., 2016*; *Pérez-González et al., 2021*). In the finite element model, a cell was represented by few quadrilateral elements and its cytosol was assumed to be viscous. The contractile forces on the cell apical, lateral, and basal surfaces were inferred from deformation of cells of the same representation (*Brodland et al., 2010*). The forces were varied in the simulation to show that the surface tension of the invaginated cells was more influential than those of surrounding cells, and that the invagination was robust against the variance in the force. In the vertex model, a cell was represented by a quadrilateral, polygon, or polyhedron with a surface tension on each edge or face and constraint on an area or volume inside. The differential surface tension was assigned based on the observation of myosin distribution in a modeled tissue and varied to investigate its contribution. These studies succeeded in reproducing the apical constriction by the increased apical surface tension, and further investigated additional components, such as differential lateral and basal surface tension, boundary condition by an extracellular matrix, and compression by the surrounding cells, to make the resultant simulated deformation more similar to the modeled actual tissue.

Along with the cell autonomous deformation with the apical constriction, in many epithelial tissues bending and invagination were found mechanically promoted by surrounding cells. For example, actomyosin was localized to a boundary between amnioserosa and lateral epidermis (*Hutson et al., 2003*; *Hayes and Solon, 2017*) or neural plate and epidermal ectoderm (*Hashimoto et al., 2015*; *Galea et al., 2017*; *Galea et al., 2018*). These supracellular cables draw arcs and merge at a midline like a canthus so that a contraction of the cable results in the decrease of the inner amnioserosa or neural plate apical surface. Similar supracellular myosin cables were found encircling salivary glands (*Röper, 2012*; *Chung et al., 2017*) and tracheal pit (*Nishimura et al., 2007*; *Ogura et al., 2018*) during their invagination. In the same way as the contracting circumferential myosin belt in a cell decreasing the cell apical surface, the circular supracellular myosin cable contraction decreases the perimeter, the radius of the circle, and the area inside the circle. Also, a planar compression was proposed to be able to drive the epithelial folding (*Hočevar Brezavšček et al., 2012*).

In this study, we revisited the driving mechanism of the apical constriction and investigated what cell physical properties were required for the cellular deformation. First we modified a cellular Potts model, which is another framework to simulate cellular tissue (*Graner and Glazier, 1992*), so that it could represent an epithelial tissue, and simulated the effect of increased apical contractility mediated by apically localized myosin. Contrary to expectations, the cellular Potts model did not reproduce tissue invagination by the apical constriction but the cells were delaminated one-by-one from edges. By analyzing a change in energy with respect to the cell shape, we found that the force to contract the cell apical surface was substantially affected by the entire cell shape. Next we considered an alternative driving mechanism for the apical constriction incorporating a cell apical surface elasticity, which might be regulated by endocytosis. It succeeded in reproducing the apical constriction and epithelial bending. In addition, the encircling supracellular myosin cable largely promoted the invagination by

the apical constriction, suggesting that too high apical surface tension may keep the epithelium apical surface flat.

## Results

### Extended cellular Potts model to simulate epithelial deformations

To model cellular and tissue deformations, we use the cellular Potts model (*Graner and Glazier, 1992*).

The model was first proposed to simulate cell sorting by assigning differential cell adhesion, which was expressed by a contact energy. In the basic cellular Potts model, cells are represented by sets of sites in a 2D lattice (*Figure 1a*), and their deformation is simulated by repeatedly updating a label on a randomly chosen site (*Figure 1b and b'*). For a site $i$, the contact energy is calculated by $\sum_{j \in N(i)} J(\sigma(i), \sigma(j))$, where $N(i)$ is a neighborhood of $i$, $\sigma(i)$ represents a cell to which $i$ belongs, and $J$ defines an affinity between the cells (*Figure 1c*). The contact energy is summed among all sites to represent a surface contact energy. Later, in various studies of cell and tissue-scale phenomena, the cellular Potts model was employed with additional elements and was extended to 3D. For example, *Fortuna et al., 2020* simulated a cell migration with material labels representing nucleus, cytoplasm, and lamellipodium within each cell. To simulate morphogenesis of an epithelial tissue with apico-basally polarized cells, the intracellular material labels were utilized to represent cell apical surface, lateral cell-cell junction, and basal surface (*Belmonte et al., 2016*; *Adhyapok et al., 2021*). Also, the cellular Potts model was combined with a finite element model so that it simulated how cells generated and were pulled by stress/strain on the extracellular matrix (ECM) (*van Oers et al., 2014*; *Rens and Merks, 2017*).

To simulate the epithelial tissue morphogenesis, we developed a new cellular Potts model with the cell apical-basal polarity. The monolayer tissue was represented in a 2D lattice in order to save on computational cost, and while the general 2D cellular Potts model cast the epithelium plane to the lattice, our model cast the tissue section perpendicular to the plane. In this model, the cell was represented by a set of sites labeled with a cell identity $\sigma$ (*Figure 1a*), a cell type, and a cytosol $\tau$ corresponding to the cell type (*Figure 1d*). The cells perimeter was partitioned automatically based on adjacency with other cells, and it was marked as apical, lateral, basal. Also, apico-lateral sites were marked as a location for the adherens junction. This cell representation also cast the vertical section of the cell. Therefore the interior area of the cell corresponded with a body of the cell, and a perimeter of the cell corresponded with the cell surface. Likewise the apical, lateral, and basal parts of the perimeter corresponded with the apical surface, cell-cell interface, and the basal surface of the cell, respectively. Outside the epithelial cells, sites were labeled as a medium (*Figure 1a*) and with materials representing an apical ECM and fluids in the apical and basal sides (*Figure 1d*).

In this study, we assumed that the cell surface tension consisted of contractility and elasticity. We modeled the contractility as a constant force to decrease the surface, but not dependent on surface width or strain. We modeled the elasticity as a force proportional to the surface strain, working to return the surface to its original width. Note that in some studies the tension and the contractility are considered as equivalent, but they are distinguished in this study. In our cellular Potts model, the surface contractility was defined by the contact energy, and expressed by

$$\mathcal{H}_{sc} = \sum_i \sum_{j \in N(i)} J(\delta(\sigma(i), \sigma(j)), \tau(i), \tau(j)), \tag{1}$$

where $\delta$ is the Kronecker delta, $\sigma$ includes the cells and medium, $\tau$ includes the cytosols, apical ECM, and fluids, and $J$ defines the contact energy between the materials $\tau(i)$ and $\tau(j)$ when the sites $i$ and $j$ are in different cells or medium, that is, $\delta(\sigma(i), \sigma(j)) = 0$. Then, the cell apical surface contractility $J_a$ was defined by the contact energy between the cytosol and the apical side medium fluid, the lateral cell-cell junction contractility $J_l$ was defined by the contact energy between the cytosols, and the basal surface contractility $J_b$ was defined by the contact energy between the cytosol and the basal side medium fluid. For the surface elasticity, a perimeter length of the 2D cell was evaluated. In the same way with the surface contact energy, a perimeter length $P$ of a cell and its apical, lateral, and basal

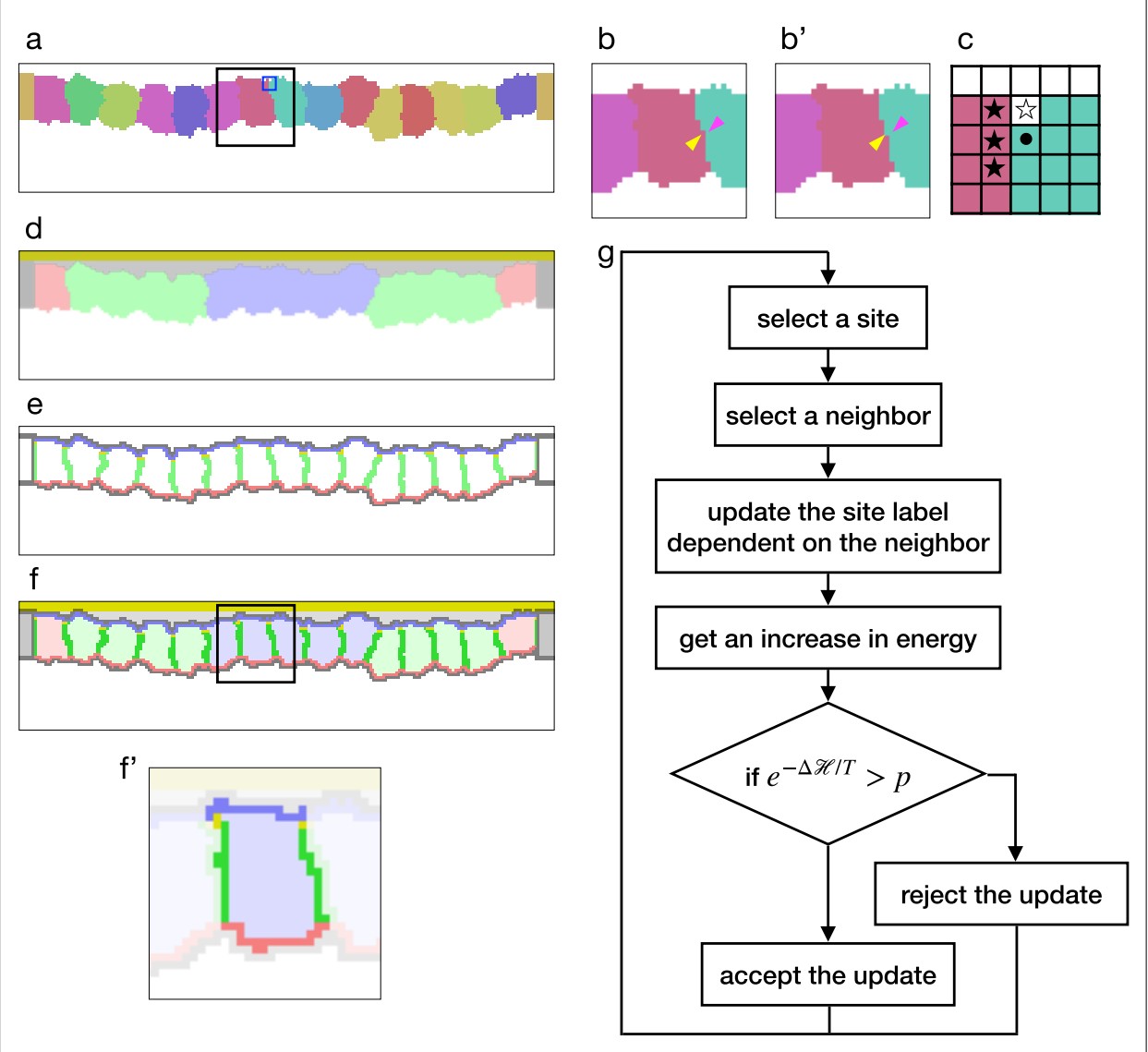

**Figure 1.** Cellular Potts model for an epithelial tissue. (**a**) Cells representation in the cellular Potts model. The cells were colored differently, and a white space represents the medium. (**b, b'**) An update of a label from (**b**) to (**b'**). The panels show an area inside a black line in (**a**). The label on a randomly selected site (yellow arrowhead) was replaced with one on a randomly selected neighboring site (magenta arrowhead). (**c**) Neighborhood of a site for the contact energy. The panel shows an area inside a blue line in (**a**). A site marked with a circle is adjacent to three sites in a different cell (black star) and a site in the medium (white star). (**d**) Material labels in the model. Four types of cytosol are colored blue, green, red, and dark gray. Inner and outer body fluid are colored white and light gray, and the apical extracellular matrix (ECM) is colored dark yellow. (**e**) Cell polarity markings. The apical, basal, lateral, and non-polarized surfaces are colored blue, red, green, and gray, respectively. The adherens junction is colored yellow. (**f, f'**) Tissue and cell representations in our model. The panel (**f'**) shows an area inside a black line in (**f**), and lightens an area around a cell. In (**f'**), the cytosol was drawn with pale blue, the cell apical, lateral, and basal surfaces were drawn with blue, green, and red, and the apical lateral sites where the adherens junction localized were colored yellow. (**g**) Algorithm of the cellular Potts model simulation.

The online version of this article includes the following figure supplement(s) for figure 1:

**Figure supplement 1.** Epithelial cell surface tension and tissue integrity.

parts $P_a$, $P_l$, and $P_b$ were calculated by summing together the adjacency of the sites to different cells, and the surface elasticity was calculated by

$$\mathcal{H}_{\mathrm{set}} = E_s(P - P_0)^2 \qquad (2)$$

for the total perimeter, or by

$$\mathcal{H}_{\text{sep}} = E_s((P_a - P_{a0})^2 + (P_l - P_{l0})^2 + (P_b - P_{b0})^2) \tag{3}$$

for the partitioned perimeters, where $E_s$ denotes a surface elastic modulus, and $P_0$, $P_{a0}$, $P_{l0}$, and $P_{b0}$ represent reference values. Note that the naming for the energy terms differs from preceding studies. For example, *Farhadifar et al., 2007* named a surface energy term expressed by a proportional function 'line tensions' and a term expressed by a quadratic function 'contractility of the cell perimeter'. In this study, however, calling the quadratic term 'contractility' would be misleading since it prevents the contraction when $P < P_0$. Therefore, we renamed the terms accordingly.

To simulate the effect of the supracellular myosin cable encircling the invaginated cells, we added a potential energy to the model. The supracellular myosin cable is formed along the adherens junctions and pulls it horizontally toward a center of the invagination in the tissue plane. The potential energy was defined by a scalar field which made a horizontal gradient decreasing toward the center, and the scalar values for the adherens junction sites were averaged in each cell:

$$\mathcal{H}_{\text{mc}} = \frac{1}{|X|} \sum_{i \in X} U(i), \tag{4}$$

where $X$ denotes a set of the adherens junction sites in the cell, and $U$ denotes the scalar field. The model also included an area constraint

$$\mathcal{H}_{\text{ac}} = \lambda(A - A_0)^2, \tag{5}$$

where $\lambda$ denotes a bulk modulus, $A$ represents an area of a cell, and $A_0$ denotes its reference value.

By combining the terms (*Equations 1–5*), a tissues total energy $\mathcal{H}$ was calculated. While the terms (*Equations 1* and *Equation 5*) were included in all simulations since they were fundamental and designed in the original cellular Potts model (*Graner and Glazier, 1992*), the other terms (*Equations 2–4*) were optional and employed only for certain conditions. The cellular Potts model simulates the cell and tissue deformation by updating the label when it decreases $\mathcal{H}$, or when it increases $\mathcal{H}$, with a probability of $e^{-\Delta\mathcal{H}/T}$, where $T$ denotes a fluctuation allowance of the system (*Figure 1g*).

Before simulating the apical constriction, we first examined parameters for a stable epithelium. The energy $\mathcal{H}$ included only the terms of the contact energy (*Equation 1*) and the area constraint (*Equation 5*), but the surface elasticity (*Equation 2*) nor (*Equation 3*) was not included, and thus the surface tension was determined by the contact energy. The term (*Equation 4*) was not included either. For a cell, its compression was determined by a balance between the pressure and the surface tension, that is, the higher surface tension would compress the cell more. The bulk modulus $\lambda$ was set 1, the lateral cell-cell junction contractility $J_l$ was varied for different cell compressions, and the apical and basal surface contractilities $J_a$ and $J_b$ were varied proportional to $J_l$. When the cell basal and apical surface tensions were equal to the lateral cell-cell junction tension, cells were rounded and disconnected from each other, and the epithelium could not maintain the integrity (*Figure 1—figure supplement 1*). Therefore, we assigned two or four times higher contact energy to the basal and apical surface than the lateral cell-cell junction in the following simulations.

## Tissue deformation by increased apical contractility simulated with cellular Potts model

To test whether the apically localized actomyosin could drive the apical constriction, we simulated the epithelial cell monolayer with increased apical surface contractility. Some preceding studies assumed that the apical myosin generated the contractile force (*Sherrard et al., 2010*; *Conte et al., 2012*; *Perez-Mockus et al., 2017*; *Pérez-González et al., 2021*), while the others assumed the myosin to generate the elastic force (*Polyakov et al., 2014*; *Inoue et al., 2016*; *Nematbakhsh et al., 2020*). It seems natural to consider that the myosin generates a force proportional to its density but not to the surface width nor the strain. For the sake of simplicity, we ignored the effect of the constriction on the apical myosin density and discuss it later. Also, an experiment with cells cultured on a micro pattern showed that the myosin activity corresponded well to the contractility, and its effect on the elasticity, if any, was minor (*Labouesse et al., 2015*). Therefore, we assigned to cells an additional apical surface contractility by setting $J_a$ larger than $J_b$ for the center pale blue cells. A simulation started from a flat monolayer of cells beneath the apical ECM and was continued until resulting deformation of cells and tissue could be evaluated for success or failure of reproducing the apical constriction.

In contrast to the simulations in the preceding studies (*Sherrard et al., 2010*; *Conte et al., 2012*; *Perez-Mockus et al., 2017*; *Pérez-González et al., 2021*), our simulations could not reproduce the apical constriction, but cells with the increased apical surface contractility were covered by surrounding cells and delaminated one by one from edges (*Figure 2a*, *Figure 2—video 1*). When the compression $A_0/\bar{A}$ and the entire surface contractility was small, the covering cells were horizontally spread and their surface was extended largely during the covering. To prevent the surface expansion, we included the surface elasticity term (*Equation 2*). It kept the cells surface perimeter in a narrower range as expected, but still the center cells were delaminated and adopted similar shapes (*Figure 2b*).

We also simulated deformation with the supracellular myosin cable but without the increased apical surface contractility. In the cross section, the shrinkage of the circular supracellular myosin cable was simulated with a move of adherens junction under the myosin cable toward the midline. Either with or without the surface elasticity, the center cells were covered by the pulled cells and delaminated (*Figure 2c*).

These results suggest that the increased apical surface contractility and the supracellular myosin cable cannot drive the apical constriction. It was not limited to specific parameters but reproducible for various contractilities. It contradicts the current model that the apical constriction is driven by the increased contractility generated by the apically localized myosin.

## Energy landscape with respect to cell shapes

To investigate the origin of the difference between the vertex model and the cellular Potts model results, we first looked into an early deformation of the cells in the cellular Potts model (*Figure 3a and b*). It was characterized by (1) that only the edge cell shrank its apical surface, (2) that inner cells remained in the columnar shape, and (3) that a lateral surface of the edge cell was curved outward. These could be explained with a balanced apical surface contractility. For the sake of simplicity, consider a system of vertices linked by edges in a line, and the edges bear differential contractilities. For a vertex, a force exerted on the vertex is a sum of the opposing contractile forces on the linking two edges, and it will be zero when the two edges bear an equal contractility. In the same way, a force exerted on an apical cell-cell junction by the apical surface contractility was a sum of them, and it would be zero when two adjacent cells bore an equal apical surface contractility. Then, a practical pulling force on the cell-cell junction by the cell apical surfaces remained only on the edge of the cells with the increased apical surface contractility (*Figure 3c–e*). The edge cell shrank its apical surface and consequently decreased its volume. The decrease in the volume increased the internal pressure due to the cell volume conservation, and a difference in the pressure between the contracting and non-contracting cells resulted in the lateral surface curvature.

Since the cellular Potts model results seemed acceptable, we next examined why the inner cells could shrink their apical surface in the vertex model by analyzing the energy with respect to cell shape. For the analysis of the cell shape in motion, we plotted a phase diagram for shapes of a single cell (*Figure 3f*). The cell shape was represented by vertices and edges linking the vertices, and its energy function was defined by a combination of the pressure and the surface contractility. While the conventional vertex model assumes only straight edges, we included also curved edges.

*Figure 3f* shows cell shapes with which the energy of the cell was at a minimum for an apical surface width and a curvature of the lateral surface.

When the cell lateral surface was restricted to be straight, a plot of energy with respect to the apical surface width was well fitted by a quadratic polynomial ($y = 0.0216x^2 - 4.8409x + 5062.8$, $R^2 = 0.9999$) (*Figure 3g and h*), indicating that the practical force to constrict the apical surface was attenuated by the deformation. Because of the attenuation, the constricting force was unbalanced between the shrunk edge cell and the inner columnar cell, and it allowed the inner cells to shrink the apical surface simultaneously. On the other hand, when the cell lateral surface was allowed to be curved and the cell changed its shape along a gradient of the energy landscape (*Figure 3g*), a plot of energy with respect to the apical surface was well fitted by a linear function ($y = -4.3135x + 5055.3$, $R^2 = 0.9992$) (*Figure 3h*), indicating that a cell could shrink its apical surface without attenuating the constricting force. When the practical constricting force remained constant, the inner cell would remain in the columnar shape.

These results indicate that the difference between the vertex model and the cellular Potts model results was due to the straight lateral surface imposed on the vertex model, a restriction that seems not

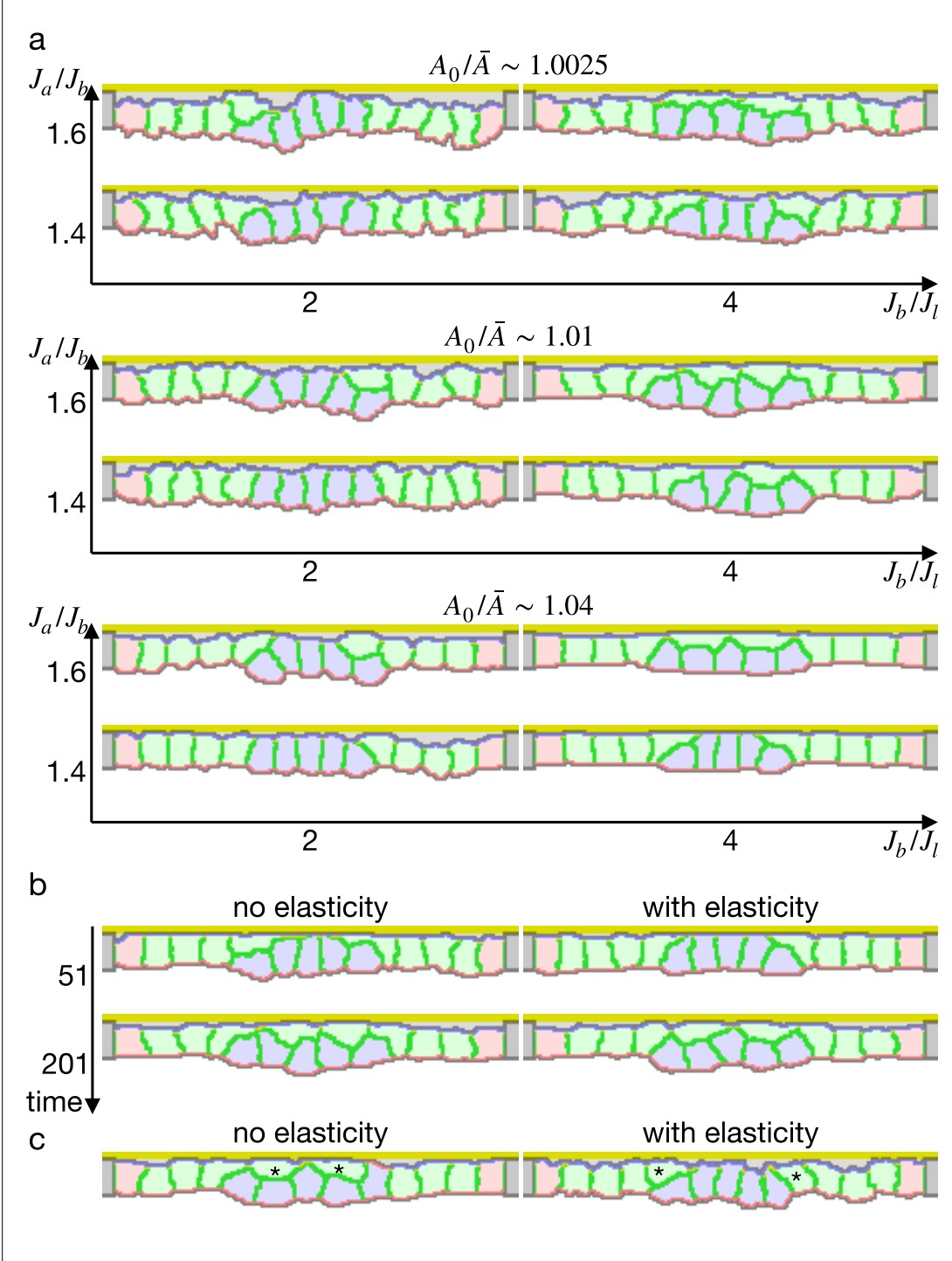

**Figure 2.** Simulations of epithelial tissue with the increased contractility. (**a**) The ratio $A_0/\bar{A}$ indicates how much cells were apico-basally compressed by the lateral cell-cell junction contractility $J_l$. The horizontal axis $J_b/J_l$ indicates a ratio between the basal surface and lateral cell-cell junction contractilities. The vertical axis $J_a/J_b$ indicates a ratio between the apical and basal surface contractilities in center pale blue five cells. The other surrounding pale red and green cells were assigned the apical surface contractility equal to the basal surface contractility. (**b**) Simulations w/o the surface elasticity, where $A_0/\bar{A} \sim 1.01$, $J_b/J_l = 4$, and $J_a/J_b = 1.6$. The vertical axis represents time (1 time/1000 updates). (**c**) Simulations of the supracellular myosin cable. Cells marked with an asterisk were assigned a potential energy on their adherens junction so that they were pulled toward the midline. $A_0/\bar{A} \sim 1.01$ and $J_b/J_l = 4$.

The online version of this article includes the following video for figure 2:

*Figure 2 continued on next page*

*Figure 2 continued*

**Figure 2—video 1.** Simulation of epithelial tissue with the increased contractility.

https://elifesciences.org/articles/93496/figures#fig2video1

natural for epithelial cells. Also, we expect that the cellular Potts model simulation with the increased apical surface contractility reproduced a phenotype of endocytosis down-regulation as discussed later.

## Apical constriction by modified apical elasticity

Since the cellular Potts model simulation indicated that the increased apical surface contractility could not drive the apical constriction, we revisited requirements for the apical constriction. It requires (1) a mechanism to simultaneously shrink the apical surface among cells in a cluster, and (2) a mechanism to keep the lateral surface straight. Actually, the condition (2) can be satisfied when the condition (1) is achieved because the simultaneous apical shrinkage will increase the pressures in the cells, but the increase will also be synchronized and thus the pressure will be balanced. As suggested by the cell shape energy analysis (*Figure 3f*), the apical shrinkage can be synchronized when the energy function and its derivative are both increasing with respect to the apical width. The energy function of the partitioned perimeters elasticity (*Equation 3*) has a such feature, and it may correspond to the stretching elasticity of cell membrane (*Helfrich, 1973*), where the apical reference value $P_{a0}$ can be modified by apical endocytosis/exocytosis. By decreasing $P_{a0}$ with the endocytosis, the cell membrane would contribute to the apical surface tension proportionally to the change $P_a - P_{a0}$.

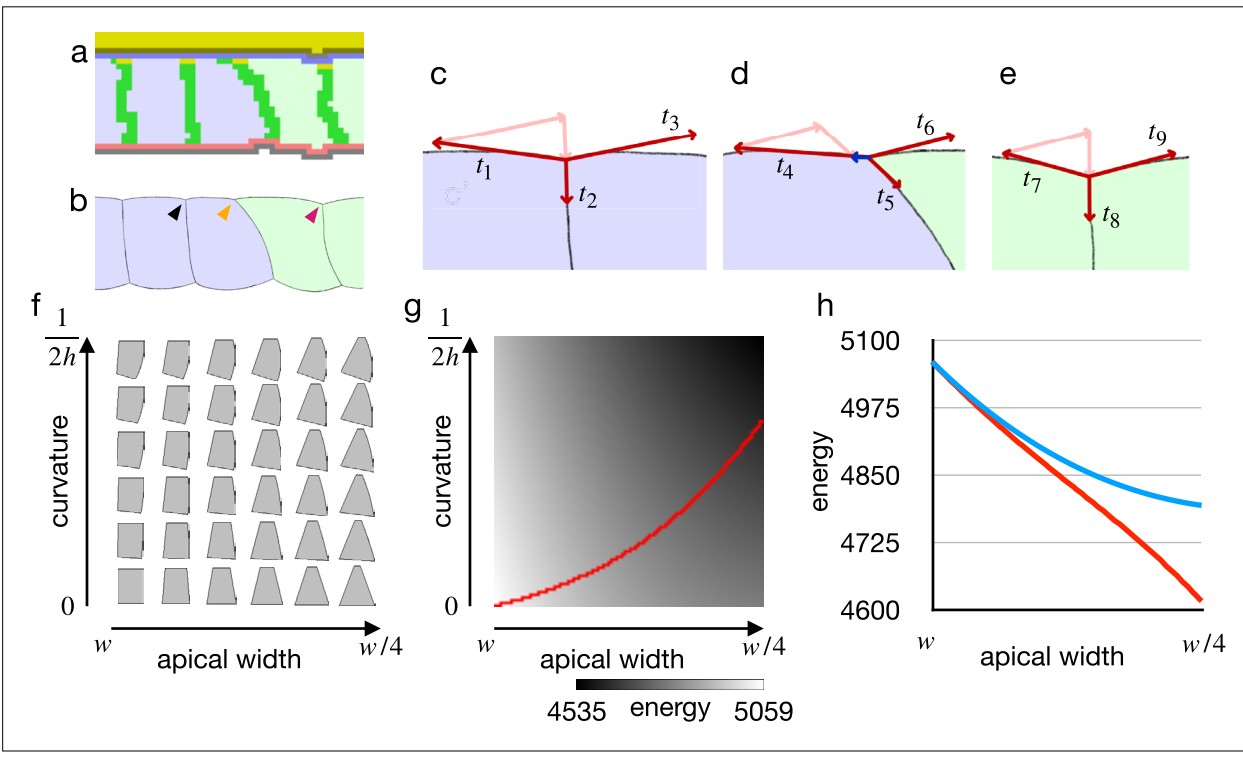

**Figure 3.** Cell shape and practical force to constrict the apical surface. (**a**) The edge cell at the early phase of the cellular Potts model simulation with the increased apical surface contractility. (**b**) Illustration of the edge cell. (**c–e**) Illustration of surface contractilities around the cell-cell junction. (**c**) Shows a junction marked by the black arrowhead in (**b**). (**d**) Shows a junction marked by the orange arrowhead in (**b**). (**e**) Shows a junction marked by the magenta arrowhead in (**b**). Vectors $t_1$-$t_9$ depict the surface contractilities exerted on the junctions. Pale pink arrows in (**c**) are the same vectors with $t_2$ and $t_3$, those in (**d**) are the same with $t_5$ and $t_6$, and those in (**e**) are the same with $t_8$ and $t_9$. Blue arrow in (**d**) depicts a sum of $t_4$, $t_5$, and $t_6$. (**f**) Phase diagram of cell shapes. For the apical width and a curvature of the right-side lateral surface, the energy of the cell is at minimum with the shape. The pressure and the surface contractility were set so that the cell took the columnar shape for the apical width $w$, apical-basal height $h$, and 0 curvature. (**g**) Energy landscape of the cell shapes for the apical width and the lateral curvature. Red line shows a path following a gradient of the energy. (**h**) Plots of energy with respect to the apical width. Blue plot shows the energy when the lateral surface was restricted to be straight. Red plot shows the energy along the path in (**e**).

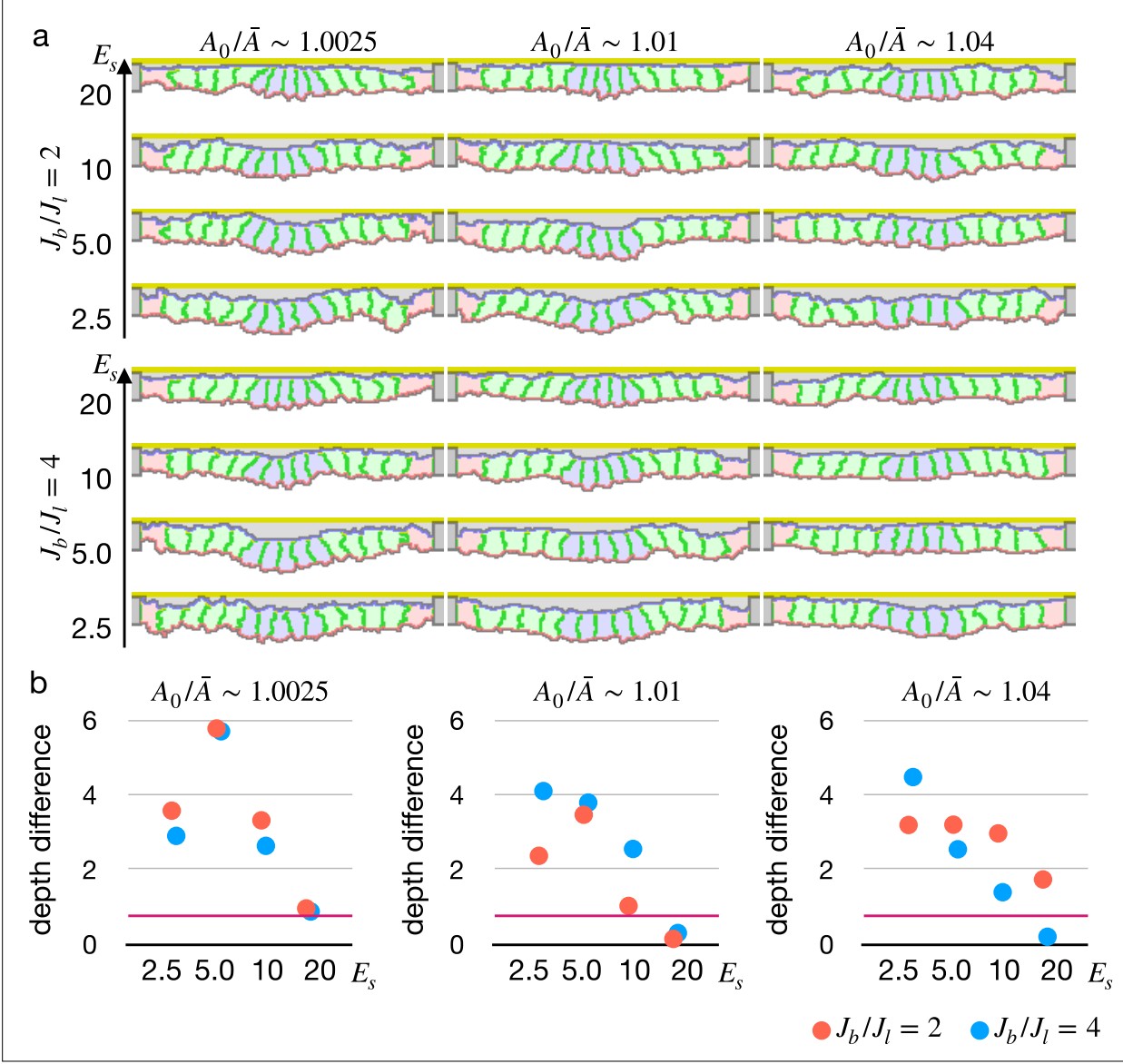

**Figure 4.** Simulations of epithelial tissue with the modified surface elasticity. (**a**) Results of the simulations. The center pale blue five cells were assigned $P_{a0} = 0$, while the others were assigned $P_{a0}$ equivalent to $P_a$ initial value. The ratio $A_0/\bar{A}$ indicates how much cells were compressed, $J_b/J_l$ indicates a ratio between the basal surface and lateral cell-cell junction contractility, and $E_s$ denotes the surface elastic modulus for the inner pale blue and green 13 cells. The edge pale red cells were assigned 0.1 times smaller surface elastic modulus than the inner cells. (**b**) Plots showing difference in distance from the apical extracellular matrix (ECM) between the constricting cells and the surrounding cells. Average distances were compared, and a larger difference indicates a deeper invagination. Magenta horizontal lines indicate 0.73, an average difference between the center cells and surrounding cells when all of the cells were assigned $P_{a0}$ equivalent to the $P_a$ initial value, as a control. Results of three simulations were averaged.

The online version of this article includes the following video and figure supplement(s) for figure 4:

**Figure supplement 1.** Simulation of epithelial tissue with the gradient contractility.

**Figure 4—video 1.** Simulation of epithelial tissue with the decreased elastic reference value.

https://elifesciences.org/articles/93496/figures#fig4video1

To validate our new model, we simulated the epithelial cell monolayer with the decreased elastic reference value (*Figure 4a*, *Figure 4—video 1*). The inner cells shrank the apical surface simultaneously, and they were deformed into the wedge shape with fluctuating but almost straight lateral surfaces, succeeding in invaginating the tissue with various parameters. Note that the simulation results were not at a steady state but in the middle of the deformation. The simulations were paused at an arbitrary time point because it was hard to assess whether the tissue was in the steady state or

not, and resultant configurations were checked to assess whether the cells were delaminated or not, and evaluated the invagination by comparing the center pale blue cells and surrounding pale green cells in an average distance to the apical ECM. Interestingly, the tissue apical surface was kept relatively flat when the surface elastic modulus was high even though the cells adopted the wedge shape (*Figure 4b*).

We also tested another model for the simultaneous apical shrinkage, a gradient contractility model (*Spahn and Reuter, 2013*; *Rauzi et al., 2015*). If the inner cells bear higher apical surface contractility than the edge cells, that inner cells may shrink their apical surface. To synchronize the apical shrinkage, the apical contractility must follow a parabola shape gradient. Even though the gradient contractility enabled the cells to shrink the apical surface simultaneously, often some of the cells shrank faster than neighbors and were delaminated by chance (*Figure 4—figure supplement 1*).

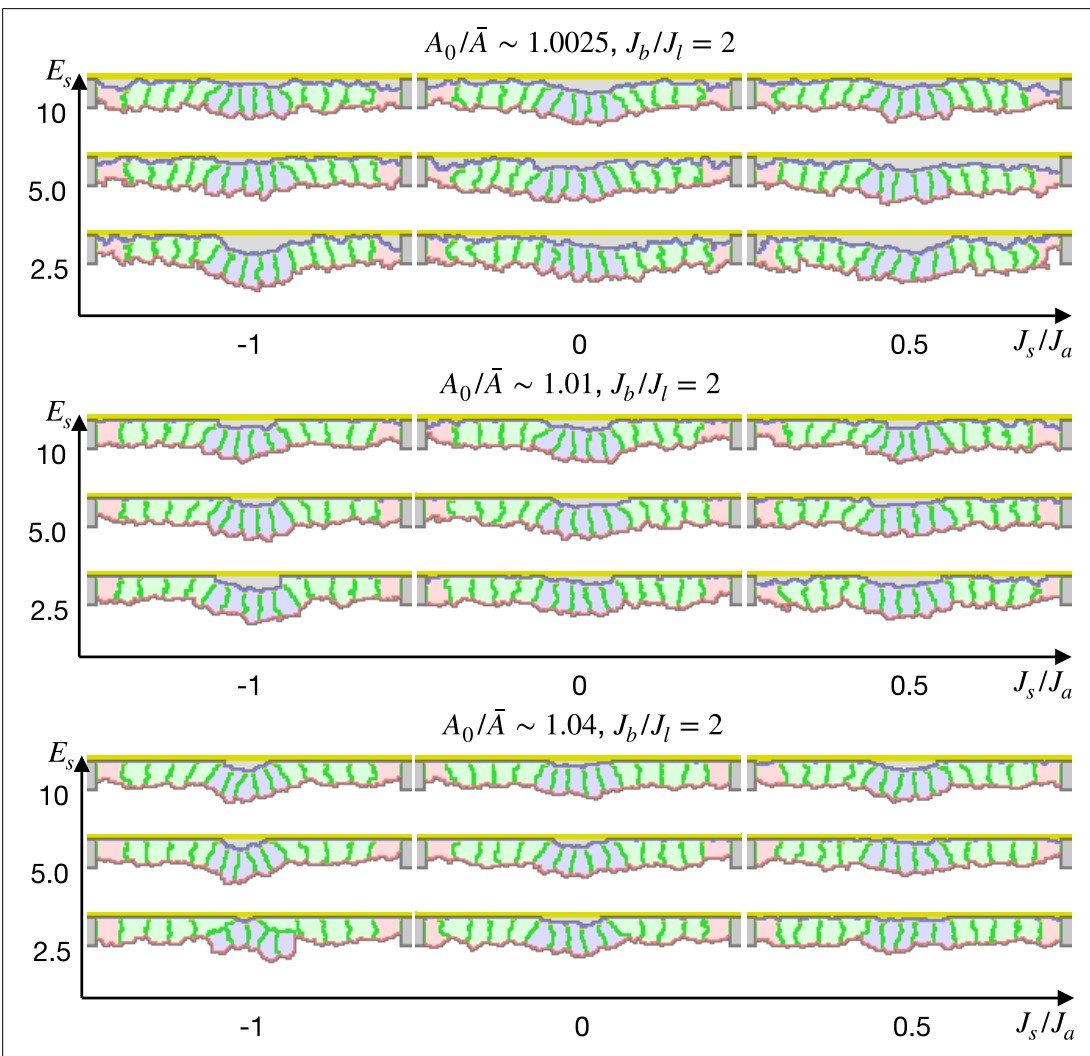

**Figure 5.** Simulation of apical constriction with the supracellular myosin cable. The center pale blue five cells were assigned $P_{a0} = 0$, and the siding pale green two cells adjacent to the center pale blue cells were assigned the potential energy on their adherens junction so that they were pulled toward the midline. The magnitude $C_r$ indicates a gradient of the potential energy, $A_0/\bar{A}$ indicates how much cells were compressed, $J_b/J_l$ indicates a ratio between the basal and lateral cell-cell junction contractility, and $E_s$ denotes the surface elastic modulus for the inner pale blue and green 13 cells. The edge pale red cells were assigned 0.1 times smaller surface elastic modulus than the inner cells.

The online version of this article includes the following figure supplement(s) for figure 5:

**Figure supplement 1.** Simulation of apical constriction with various cell heights.

**Figure supplement 2.** Simulation of apical constriction by the elasticity remodeling with cell-extracellular matrix (ECM) adhesion.

## Variation in configuration and resultant deformations

Next we simulated the apical constriction by the elasticity remodeling with various conditions. The apical constriction could bend the tissue with thinner and thicker cells (*Figure 5—figure supplement 1*), demonstrating that the model was robust against cell shapes. When the surrounding cells were assigned an affinity to the apical ECM, the cells were spread on the ECM and pushed the center cells toward the midline, and the invagination was slightly promoted (*Figure 5—figure supplement 2*).

We also simulated the apical constriction by the elasticity remodeling with additional supracellular myosin cable (*Figure 5*). The edge pale green cells adjacent to the center pale blue cells were assigned the potential energy so that their adherens junctions were pulled toward the midline (*Equation 4*). Surprisingly, the tissue invagination was largely promoted by the supracellular myosin cable with a wide range of the pulling force and also wide ranges of other parameters. Note that the supracellular myosin cable alone could not reproduce the apical constriction (*Figure 2c*), and that with some parameters the modified cell surface elasticity kept the tissue almost flat (*Figure 4*). However, combining both the supracellular myosin cable and the cell surface elasticity made a sharp bending when the pulling force acting on the adherens junction was sufficiently high.

## Inferred pressure distribution around *Drosophila* embryo tracheal pit

In the new model, we proposed that the simultaneous increase in the pressure would keep the cell lateral cell-cell junction straight. However, it is also possible that other mechanisms may keep the lateral cell-cell junction straight, for example, by decreasing cell volume so that the pressure is constant and balanced among the cells. To check if the cell pressure was increased in an actual tissue, we implemented a geometrical tension inference around an invaginated tracheal pit of *Drosophila* embryo. The geometrical tension inference is a method to measure relative junctional tensions and cell pressures from shapes of the cells (*Roffay et al., 2021*), and we extended 2D Bayesian method (*Ishihara and Sugimura, 2012*) to 3D for the measurement of invaginated tissue.

The tracheal pit is formed in a tracheal placode by the apical constriction (*Nishimura et al., 2007*; *Kondo and Hayashi, 2013*). An initially flat apical surface of the tissue was bent to make the pit (*Figure 6a*). During the process, there was no obvious change in the distribution pattern of cellular junctional tension inside and around the tracheal pit (*Figure 6b*). On the other hand, the apical surface area decreased and the relative pressure increased among the invaginated cells (*Figure 6c–e*). Note that variances of the apical area and the relative pressure did not increase (*Figure 6d and e*), suggesting that an actual pressure value was also increased simultaneously by the cell deformation.

These results were consistent with our model, indicating that the lateral cell-cell junction of the tracheal pit cell was kept straight by the increasing but balanced pressure.

## Discussion

In this study, we demonstrated that the increased apical surface contractility could not drive the apical constriction and proposed the alternative driving model with the apical surface elasticity remodeling. When the cells were assigned a high apical surface contractility, the cells became rounded and delaminated one by one from edges in the cellular Potts model simulation. These simulation results disagree with the conventional model of the apical constriction (*Martin and Goldstein, 2014*; *Takeichi, 2014*; *Martin, 2020*) and its vertex model simulation results. However, the vertex model simulation was substantially affected by the straight lateral surface (*Figure 3d–f*), which was not natural for the epithelial cells. An edge in the vertex model can be bent by interpolating vertices or can be represented with an arc of circle (*Brakke, 1992*). Even in cases where vertex models were extended to allow bent lateral surfaces, the model still limited cell rearrangement and neighbor changes (*Pérez-González et al., 2021*), limiting the cell delamination. Thus the difference in simulation results between the models could be due to whether the cell rearrangement was included or not. However, it is not clear how the absence of the cell rearrangement affected cell behaviors in the simulation, and it shall be studied in future. In contrast to the vertex model, the cellular Potts model included the curved cell surface and the cell rearrangement innately, it elucidated the importance of those factors. For the apical constriction, we considered the modified apical surface elasticity as a plausible underlying mechanism, where the synchronized decrease in cell volume and increase in pressure balances the pressure between the cells and keeps the lateral cell-cell interface straight. The requirement of simultaneous

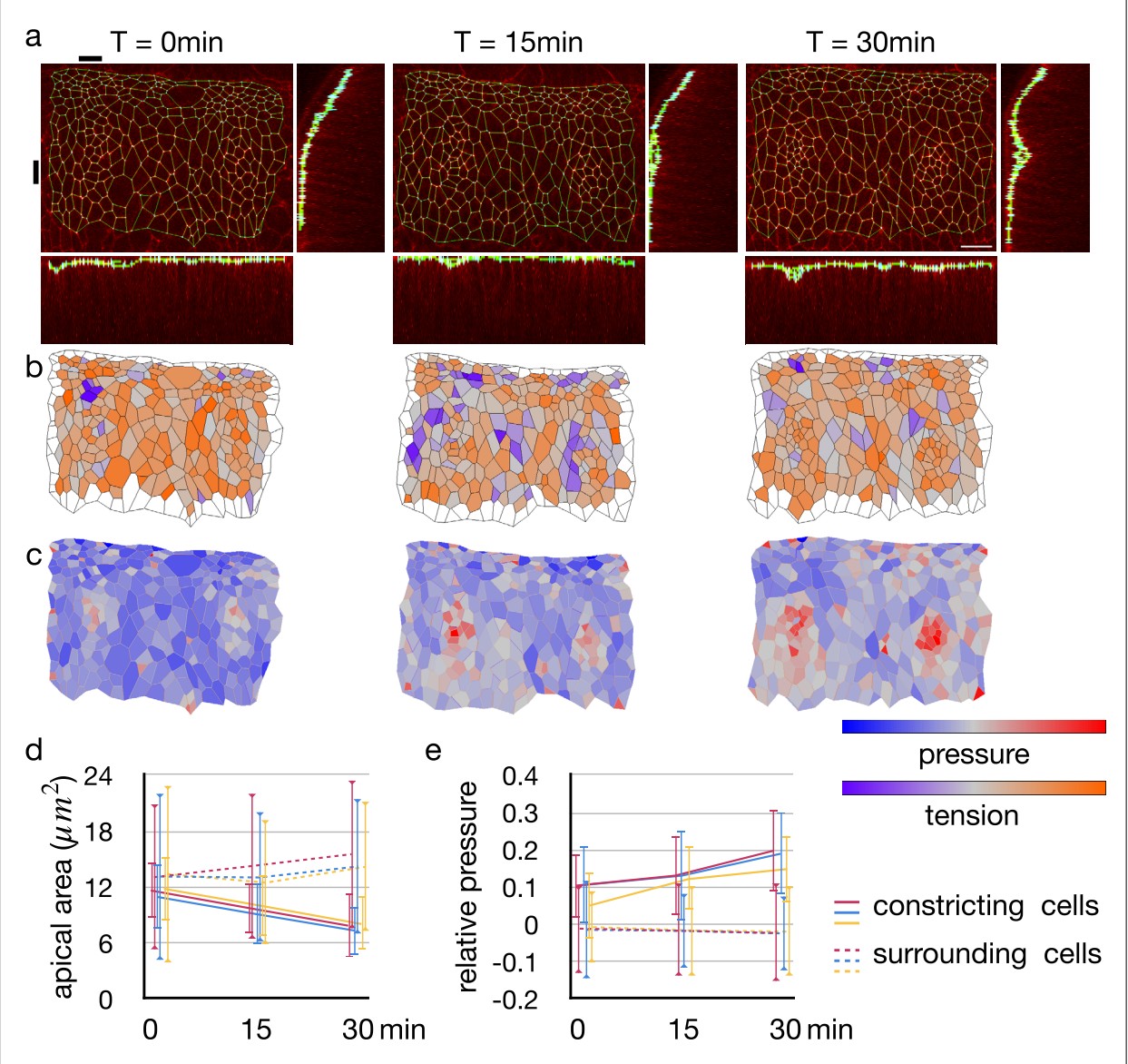

**Figure 6.** Change in junctional tension and cell pressure distribution during tracheal pit invagination. (**a**) Vertices and edges representation of adherens junction inside and around *Drosophila* embryo tracheal pit. The panels show two tracheal pits from the beginning of the invagination (0 min) and after 15 and 30 minutes. Black bars on top and at left side indicates positions of the y-z and x-z slices. (**b**) Heat maps showing average junctional tensions in each cell at the three time points. The relative junctional tensions were averaged weighted with the edge lengths for each cell. (**c**) Heat maps showing relative cell pressure in blue-red and relative junctional tension in purple-orange at the three time points. (**d**) Change in apical surface area among the invaginated constricting cells (lines) and surrounding cells (dotted lines). Colors indicate different embryos, and the values were averaged in each embryo (n = 26, 34, and 33 for red, blue and yerllow lines, and n > 238 for dotted lines). Error bars indicate SDs. (**e**) Change in relative cell pressure among the invaginated constricting cells (lines) and surrounding cells (dotted lines). Colors indicate different embryos, and the values were averaged in each embryo (n were same with above). Error bars indicate SDs. A scale bar in a represents 10 nm.

contraction is also consistent with the non-autonomous disruption of apical constriction (*Guglielmi et al., 2015*; *Galea et al., 2021*). With the elasticity-based apical constriction model, the cellular Potts model reproduced a tissue invagination. It succeeded with a wide range of parameter values, indicating a robustness of the model. Note that the vertex model could also be extended to incorporate the curved edges and rearrangement of the cells by specifically programming them and would reproduce the cell delamination. That is, we could find the importance of the balanced pressure because the cellular Potts model intrinscally included a high degree of freedom for the cell shape, the cell rearrangement, and the fluctuation. Also, we confirmed the increase in pressure among invaginated

cells in *Drosophila* embryo tracheal pit. In the preceding studies, the apically localized myosin was assumed to generate either the contractile force (*Sherrard et al., 2010*; *Conte et al., 2012*; *Perez-Mockus et al., 2017*; *Pérez-González et al., 2021*) or the elastic force (*Polyakov et al., 2014*; *Inoue et al., 2016*; *Nematbakhsh et al., 2020*). However, the limited cell shape in the vertex model made them similar in terms of the energy change during the apical constriction, that is, the effective force to decrease the apical surface. In this study, we showed that the contractile force and the elastic force differently deformed the cells and tissue, and demonstrated why and how the elasticity was important for the apical constriction.

Interestingly, the tissue apical surface remained relatively flat when the cells were assigned higher surface elastic modulus. Though it might seem contradicting that the surface elasticity was the dominant factor for the apical constriction, it could be explained that a too high surface tension will keep the apical surface short and thus straight as a result.

In our model, for the sake of simplicity, we ignored an effect of the constriction on the apical myosin density. If we presumed that the apical myosin would be condensed by the shrinkage of the apical surface, it would increase the apical tension in the shrinking cell and is expected to promote the cell delamination further. Therefore, it would not change the results.

If the apical surface elasticity was the determinant factor for the apical constriction, then how does the apical surface contractility generated by the myosin contribute to it, and how is the apical surface elasticity regulated by molecules? In many tissues, the apical myosin was found not permanently active but its flow, activation, and resultant apical surface contraction was pulsed, followed by a partial relaxation (*Martin et al., 2009*; *Solon et al., 2009*; *Martin et al., 2010*; *Mason et al., 2013*; *Booth et al., 2014*; *Kerridge et al., 2016*; *Mason et al., 2016*; *Krueger et al., 2020*; *Martin and Goldstein, 2014*; *Miao and Blankenship, 2020*). The contraction and partial relaxation is iterated to decrease the apical surface progressively, and it is called ratchet model or ratcheting mechanism.

While it is yet to be clarified how the ratcheting prevents the apical surface from expanding back to its original width during the relaxation, some studies pointed out an involvement of the endocytosis in the apical constriction (*Lee and Harland, 2010*; *Mateus et al., 2011*; *Ossipova et al., 2014*; *Ossipova et al., 2015*; *Le and Chung, 2021*), and especially in its ratcheting (*Miao et al., 2019*; *Kowalczyk et al., 2021*; *Chen and He, 2022*). When the apical endocytosis was impaired but the myosin activity was retained, cells exhibited the pulsed apical contraction but fully turned back to their original shape in the following relaxation. Interestingly, *Miao et al., 2019* downregulated endocytosis in a *Drosophila* gastrula and observed the apical contraction in the edge cells of a mesoderm which would be invaginated by the apical constriction in the wild type embryo. This edge-specific contraction resembles the result of simulations with the increased apical surface contractility (*Figure 2*), and the retained myosin activity might be consistent with the increased contractility. Together, the myosin surely generates the pulsed contractility and also might be constantly generating contractility at some magnitude. Then, how do the myosin and its pulsed and constant contractility contribute to the ratcheting and the apical constriction? Although most of the preceding studies supposed that the endocytosis regulated the ratcheting indirectly by facilitating apical actomyosin network remodeling, here we would propose that the cell membrane itself increases the apical surface tension by changing the reference value. Since the cell membrane is an elastic sheet, it bears the surface tension when extended, and the surface tension magnitude is proportional to the change from the reference value. The reference value is an area of the membrane when the membrane is not stretched nor compressed, and it can be decreased by the endocytosis. It might be also possible that the remodeling of the apical actomyosin ratchets the relaxation, as actin filament linker βH-spectrin was also reported regulating the ratcheting (*Krueger et al., 2020*). However, a resultant apical surface tension must be increasing with respect to the apical area, and an accountable mechanism is not known for the actomyosin network for such ratcheting and physical property.

In summary, we propose that (1) the actomyosin generates pulsed contractility, and the cell apical surface is contracted temporally; (2) the contraction will be fully relaxed without the endocytosis; (3) the endocytosis decreases the reference value of the apical surface elasticity during the pulsed contraction; and (4) after the pulsed contraction and endocytosis, the cell apical surface is partially relaxed (*Figure 7a*).

From the model of pulsed contractility and modified elasticity, we got expectations as described below. When the apical myosin is constantly active as in the conventional model, the increased

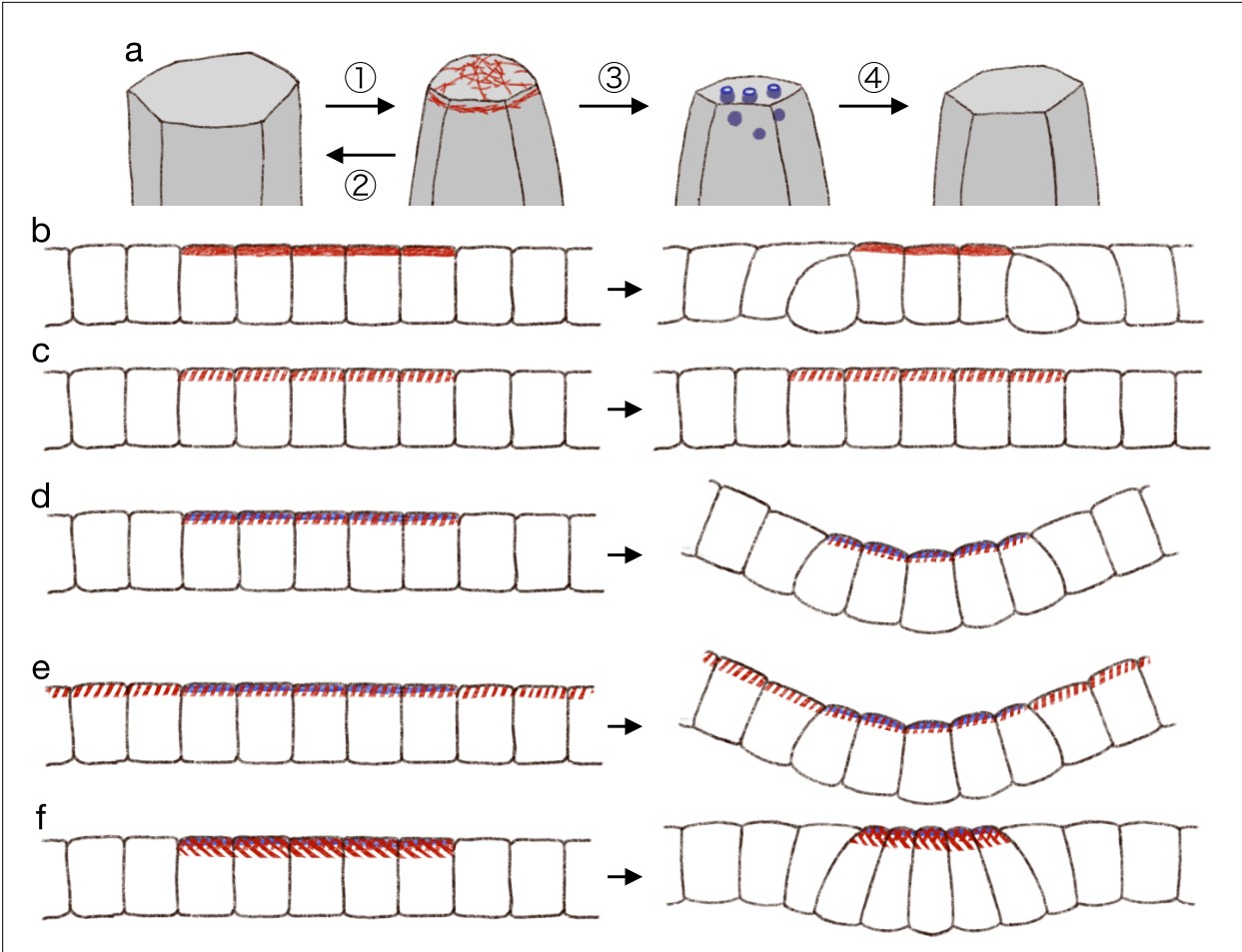

**Figure 7.** Hypothetical model of endocytosis-based apical constriction. (**a**) Flow diagram of the ratcheting by endocytosis. The cell apical surface was contracted by the pulsed myosin activation (1). Without the endocytosis, the apical surface would be fully relaxed (2). By the endocytosis, the apical surface reference value was decreased (3). Because of the modified reference value, the cell apical surface was partially relaxed (4). (**b**) Expected deformation by the increased apical surface contractility. (**c**) Expected deformation by the sporadic apical surface contractility. (**d**) Expected deformation by the patterned sporadic contractility and endocytosis. (**e**) Expected deformation by the general sporadic contractility and the patterned endocytosis. (**f**) Expected deformation with the increased apical surface tension.

contractility is balanced among the cells and makes the edge cells delaminated (*Figure 7b*). Therefore, the myosin activation must be pulsed and sporadic. When the myosin activity is pulsed and sporadic but not accompanied by the surface elasticity modification, the temporal contraction is fully relaxed and the tissue remains in the same shape (*Figure 7c*). When the sporadic pulse of myosin activation is followed by the endocytosis, the cells undergo the simultaneous apical contraction, and then the pressure is balanced among the cells, keeping the cell lateral cell-cell junction straight and deforming the cells into the wedge shape (*Figure 7d*). In the actual tissue, the apical localization of the myosin and regulators of the endocytosis were both patterned. However, it may be possible in theory that the myosin activation is not patterned but is pulsed across the entire tissue, while the endocytosis is patterned, and the tissue will be bent at the site of the increased endocytotic activity (*Figure 7e*). Another interesting notion was that too high myosin activation would prevent the apical constriction. Even if the apical contraction is synchronized and the lateral surface is kept straight so that the cells are deformed into the wedge shape, the too high apical surface tension will try to make the apical surface as short as possible, and the apical surface will be kept flat (*Figure 7f*). On the other hand, mechanisms which does not generate the apical surface tension, such like the zippering (*Bahri et al., 2010*; *Hashimoto et al., 2015*; *Galea et al., 2017*) or the supracellular myosin cable encircling the contracting cells (*Nishimura et al., 2007*; *Röper, 2012*; *Chung et al., 2017*), may promote the deep invagination. Note that we also demonstrated that the supracellular contractile ring alone could

not drive the tissue invagination (*Figure 2c*). How actual individual tissues accomplish the synchronized apical contraction would be clarified by examining the physical property and dynamics of the cell membrane, the apical actomyosin network, the adherens junction, and the tight junction.

In conclusion, we elucidated how distinctively the surface contractility and the surface elasticity would regulate tissue morphogenesis. Our simulation results were consistent with the studies of the endocytosis in apical constriction, and suggested why and how the endocytosis and the ratcheting was required for the apical constriction. The depth, curvature, and speed of the invagination might be influenced by the cell shape, configuration, and parameters, and how each condition contributes to the invagination shall be studied in future.

## Methods
### Cellular Potts model with subcellular components

To simulate epithelial cell and tissue deformation, we developed a cellular Potts model with subcellular components, subcellular locations, and a potential energy. The cellular Potts model is a stochastic simulation framework representing a cell with a set of sites in a lattice, and the deformation is simulated by updating a label on the site in Metropolis-style. In the update, a change in energy $\Delta \mathcal{H}$ is calculated between the original and a proposed new configuration, and accepted if $\Delta \mathcal{H} \leq 0$ or with probability $e^{-\Delta \mathcal{H}/T}$ if $\Delta \mathcal{H} > 0$, where $T$ represents a system temperature. With the subcellular components, the energy $\mathcal{H}$ is defined by a contact energy between the subcellular components inside and across cells, and an area constraint of the cell. The contact energy is defined by

$$\mathcal{H}_{\mathrm{sc}} = \sum_i \sum_{j \in N(i)} J(\delta(\sigma(i), \sigma(j)), \tau(i), \tau(j)),$$

(6)

where each of $i$ and $j$ denotes a site, $N(i)$ is a neighborhood of $i$, $\sigma(i)$ represents a cell to which $i$ belongs, $\tau(i)$ represents a subcellular component at $i$, $\delta$ is the Kronecker delta, and $J$ define the affinity between the subcellular components either inside or across cells. The area constraint is defined by

$$\mathcal{H}_{\mathrm{ac}} = \sum_\sigma \lambda (A(\sigma) - A_0(\sigma))^2,$$

(7)

where $\sigma$ denotes a cell, $\lambda$ denotes a bulk modulus, $A(\sigma)$ represents an area of $\sigma$, and $A_0(\sigma)$ denotes a reference value for $\sigma$.

A ratio $A_0/\bar{A}$ is a parameter outside the simulation, and $A_0(\sigma)$ for a cell $\sigma$ with a width $w$ and a height $h$ at an initial configuration is set $whA_0/\bar{A}$. The contact energy between cytosols across cells, which defines the cell lateral surface contractility $J_l$, is also dependent on $A_0/\bar{A}$ and set $\lambda w^2 h(A_0/\bar{A} - 1)$. The apical surface contractility $J_a$ and the basal surface contractility $J_b$ are defined by the contact energy for the cytosol to *outer body fluid* and *inner body fluid* across cells, respectively, and set proportional to $J_l$.

In the update, first a site $i$ is randomly selected, and then another site $j$ is randomly selected from the neighborhood $N(i)$. New labels representing cell and subcellular component for the site $i$ are determined based on the labels on $i$ and $j$. For example, if the site $i$ was labeled with a cell $\sigma_1$ and $j$ was labeled as *medium* for cell identity and with *apical ECM* for the subcellular component, the new label for $i$ was *medium* and *outer body fluid*. If the site $i$ was labeled with *medium* and *apical ECM*, then the update was rejected so that a cell would not intrude into the apical ECM. The update is iterated for an arbitrary duration, and results were visually checked if the tissue integrity was retained, the cells were delaminated, or the tissue was bent or invaginated.

### Cellular Potts model with partial surface elasticity

To simulate the differential physical properties of the apical, lateral, and basal surfaces, the subcellular locations are marked automatically, and the marking is updated during the simulation. In each cell, sites adjacent to different cells but not to the medium are marked as lateral. At the initial configuration, sites adjacent to the apical ECM are marked as apical, and during the simulation, sites adjacent to medium and other apical sites in the same cell are marked as apical. At the initial configuration and during the simulation, sites adjacent to medium and not marked as apical are marked as basal. Therefore, once a cell is delaminated and loses its apical surface, afterward all sites in the cell adjacent

to the medium are marked as basal even if it is adjacent to the apical ECM or the outer body fluid. In addition, sites are marked as adherens junction if they are not apical, adjacent to an apical site of any cell, and adjacent to a lateral site of different cell.

For the surface elasticity, the total, apical, lateral, and basal perimeter lengths of a cell $\sigma$, $P_a(\sigma)$, $P_l(\sigma)$, and $P_b(\sigma)$ are calculated in the same way with the contact energy, that is, by summing numbers of adjacent sites in different cells for all sites, all sites marked apical, lateral, and basal in $\sigma$ respectively. The surface elasticity is calculated by

$$\mathcal{H}_{\text{set}} = \sum_{\sigma} E_{s\sigma}(P(\sigma) - P_0(\sigma))^2 \tag{8}$$

for the total perimeter, or by

$$\mathcal{H}_{\text{sep}} = \sum_{\sigma} E_{s\sigma}((P_a(\sigma) - P_{a0}(\sigma))^2 + (P_l(\sigma) - P_{l0}(\sigma))^2 + (P_b(\sigma) - P_{b0}(\sigma))^2) \tag{9}$$

for the partitioned perimeters, where $E_{s\sigma}$ denotes a surface elastic modulus of the cell $\sigma$ and $P_0(\sigma)$, $P_{a0}(\sigma)$, $P_{l0}(\sigma)$, and $P_{b0}(\sigma)$ represents reference values.

## Cellular Potts model with potential energy

To simulate an effect of supracellular myosin cable encircling the invaginated cells, we assigned a potential energy to the adherens junction. In 3D, tension on a circular actomyosin cable would shrink the circle, and the shrinkage would pull the cable toward the center of the circle. In 2D cross section, the cable is pulled horizontally toward the middle line. Quantitatively, the cable is pulled by $\gamma/R$, where $\gamma$ is the tension and $R$ is the radius of the cable curvature as determined from the Young–Laplace equation. The pulling force was simulated by a gradient of the potential energy decreasing toward the middle line, constant in each half side and taking V shape. The potential energy was defined by a scalar field, and the scalar values corresponding to the adherens junction sites were averaged in each cell by

$$\mathcal{H}_{\text{mc}} = \sum_{\sigma} \frac{1}{|X_\sigma|} \sum_{i \in X_\sigma} U(i), \tag{10}$$

where $X_\sigma$ denotes a set of the adherens junction sites in the cell $\sigma$, and $U$ denotes the scalar field. When the adherens junctions sites moved toward the middle line, $\mathcal{H}_{\text{mc}}$ would decrease, and the movement would be accepted if the increase of other terms in $\mathcal{H}$ was smaller than the decrease. The pulling force $\gamma/R$ is equivalent to a rate of the gradient change.

## Implementation of the simulations

The simulations were implemented by MATLAB custom scripts which are available at GitHub (https://doi.org/10.5281/zenodo.8260142; *Yamashita, 2023a*).

## Parameters for the simulations

The parameters were varied in a range, and the figures showed simulations with parameter values within a sub-range so that the results showed both success and failure in a development of interest. In each figure, snapshots of the simulations show deformation by the same time length unless specified. The ratio $A_0/\bar{A}$ indicated a balance between the pressure and an overall surface contractility, and its value 1.0025 corresponded with a low surface contractility, 1.04 corresponded with a high surface contractility, and 1.01 corresponded with a middle case. The reference value $A_0$ was set a product of $A$ initial value and $A_0/\bar{A}$.

*Table 1* shows the parameters for the simulations shown in *Figure 1—figure supplement 1*.

In *Table 1*, $w$ and $h$ indicate the width and height of a cell in the initial configuration, cell type color corresponded with that of cytosol in the figures, $J_a$ indicate the contact energy between the cytosol and the outer body fluid or the apical ECM, $J_l$ indicate the contact energy between the cytosol and the cytosol in a different cell, $J_b$ indicate the contact energy between the cytosol and the inner body fluid, and $T$ denotes the fluctuation allowance. The variable $x$ varied from 1 to 8.

**Table 1.** Parameters for epithelial cell surface tension and tissue integrity.

| Set# | $w$ | $h$ | Cell type | $A_0/\bar{A}$ | $J_a$, $J_b$ | $J_l$ | $\lambda$ | $E_s$ | $T$ |
|---|---|---|---|---|---|---|---|---|---|
| | | | Red cell | 1.0025 | $7.605 \times x$ | 7.605 | 1 | 0 | |
| (1) | 13 | 18 | Green cell | 1.0025 | $7.605 \times x$ | 7.605 | 1 | 0 | 30 |
| | | | Red cell | 1.01 | $30.42 \times x$ | 30.42 | 1 | 0 | |
| (2) | 13 | 18 | Green cell | 1.01 | $30.42 \times x$ | 30.42 | 1 | 0 | 60 |
| | | | Red cell | 1.04 | $121.68 \times x$ | 121.68 | 1 | 0 | |
| (3) | 13 | 18 | Green cell | 1.04 | $121.68 \times x$ | 121.68 | 1 | 0 | 120 |

*Table 2* shows the parameters for the simulations shown in *Figure 2*.

**Table 2.** Parameters for increased apical contractility.

| Set# | $w$ | $h$ | Cell type | $A_0/\bar{A}$ | $J_a$ | $J_l$ | $J_b$ | $\lambda$ | $E_s$ | $T$ |
|---|---|---|---|---|---|---|---|---|---|---|
| | | | Red cell | 1.0025 | 15.21 | 7.605 | 15.21 | 1 | 0 | |
| | | | Green cell | 1.0025 | 15.21 | 7.605 | 15.21 | 1 | 0 | |
| (4) | 13 | 18 | Blue cell | 1.0025 | $15.21 \times x$ | 7.605 | 15.21 | 1 | 0 | 30 |
| | | | Red cell | 1.0025 | 30.42 | 7.605 | 30.42 | 1 | 0 | |
| | | | Green cell | 1.0025 | 30.42 | 7.605 | 30.42 | 1 | 0 | |
| (5) | 13 | 18 | Blue cell | 1.0025 | $30.42 \times x$ | 7.605 | 30.42 | 1 | 0 | 30 |
| | | | Red cell | 1.01 | 60.84 | 30.42 | 60.84 | 1 | 0 | |
| | | | Green cell | 1.01 | 60.84 | 30.42 | 60.84 | 1 | 0 | |
| (6) | 13 | 18 | Blue cell | 1.01 | $60.84 \times x$ | 30.42 | 60.84 | 1 | 0 | 60 |
| | | | Red cell | 1.01 | 121.68 | 30.42 | 121.68 | 1 | 0 | |
| | | | Green cell | 1.01 | 121.68 | 30.42 | 121.68 | 1 | 0 | |
| (7) | 13 | 18 | Blue cell | 1.01 | $121.68 \times x$ | 30.42 | 121.68 | 1 | 0 | 60 |
| | | | Red cell | 1.04 | 243.36 | 121.68 | 243.36 | 1 | 0 | |
| | | | Green cell | 1.04 | 243.36 | 121.68 | 243.36 | 1 | 0 | |
| (8) | 13 | 18 | Blue cell | 1.04 | $243.36 \times x$ | 121.68 | 243.36 | 1 | 0 | 120 |
| | | | Red cell | 1.04 | 486.72 | 121.68 | 486.72 | 1 | 0 | |
| | | | Green cell | 1.04 | 486.72 | 121.68 | 486.72 | 1 | 0 | |
| (9) | 13 | 18 | Blue cell | 1.04 | $486.72 \times x$ | 121.68 | 486.72 | 1 | 0 | 120 |
| | | | Red cell | 1.01 | 121.68 | 30.42 | 121.68 | 1 | 0.01 | |
| | | | Green cell | 1.01 | 121.68 | 30.42 | 121.68 | 1 | 0.1 | |
| (10) | 13 | 18 | Blue cell | 1.01 | 194.688 | 30.42 | 121.68 | 1 | 0.1 | 60 |
| | | | Red cell | 1.01 | 121.68 | 30.42 | 121.68 | 1 | 0 | |
| | | | Green cell | 1.01 | 121.68 | 30.42 | 121.68 | 1 | 0 | |
| (11) | 13 | 18 | Blue cell | 1.01 | 121.68 | 30.42 | 121.68 | 1 | 0 | 60 |
| | | | Red cell | 1.01 | 121.68 | 30.42 | 121.68 | 1 | 0.25 | |
| | | | Green cell | 1.01 | 121.68 | 30.42 | 121.68 | 1 | 2.5 | |
| (12) | 13 | 18 | Blue cell | 1.01 | 121.68 | 30.42 | 121.68 | 1 | 2.5 | 60 |

Parameter labels represent the same ones with *Table 1*, and the variable $x$ varied from 1.4 to 1.6. The set# (4), (6), and (8) correspond to $J_b/J_l = 2$, and the set# (5), (7), and (9) correspond to $J_b/J_l = 4$

**Table 3.** Parameters for modified surface elasticity.

| Set# | $w$ | $h$ | Cell type | $A_0/\bar{A}$ | $J_a$ $/J_b$ | $J_l$ | $\lambda$ | $E_s$ | $T$ |
|---|---|---|---|---|---|---|---|---|---|
| | | | Red cell | 1.0025 | 15.21/30.42 | 7.605 | 1 | 0.25 | |
| | | | Green cell | 1.0025 | 15.21/30.42 | 7.605 | 1 | 2.5 | |
| (13) | 13 | 18 | Blue cell | 1.0025 | 15.21/30.42 | 7.605 | 1 | 2.5 | 240 |
| | | | Red cell | 1.0025 | 15.21/30.42 | 7.605 | 1 | 0.5 | |
| | | | Green cell | 1.0025 | 15.21/30.42 | 7.605 | 1 | 5.0 | |
| (14) | 13 | 18 | Blue cell | 1.0025 | 15.21/30.42 | 7.605 | 1 | 5.0 | 240 |
| | | | Red cell | 1.0025 | 15.21/30.42 | 7.605 | 1 | 1 | |
| | | | Green cell | 1.0025 | 15.21/30.42 | 7.605 | 1 | 10 | |
| (15) | 13 | 18 | Blue cell | 1.0025 | 15.21/30.42 | 7.605 | 1 | 10 | 360 |
| | | | Red cell | 1.0025 | 15.21/30.42 | 7.605 | 1 | 2 | |
| | | | Green cell | 1.0025 | 15.21/30.42 | 7.605 | 1 | 20 | |
| (16) | 13 | 18 | Blue cell | 1.0025 | 15.21/30.42 | 7.605 | 1 | 20 | 480 |
| | | | Red cell | 1.01 | 60.84/121.68 | 30.42 | 1 | 0.25 | |
| | | | Green cell | 1.01 | 60.84/121.68 | 30.42 | 1 | 2.5 | |
| (17) | 13 | 18 | Blue cell | 1.01 | 60.84/121.68 | 30.42 | 1 | 2.5 | 240 |
| | | | Red cell | 1.01 | 60.84/121.68 | 30.42 | 1 | 0.5 | |
| | | | Green cell | 1.01 | 60.84/121.68 | 30.42 | 1 | 5.0 | |
| (18) | 13 | 18 | Blue cell | 1.01 | 60.84/121.68 | 30.42 | 1 | 5.0 | 240 |
| | | | Red cell | 1.01 | 60.84/121.68 | 30.42 | 1 | 1 | |
| | | | Green cell | 1.01 | 60.84/121.68 | 30.42 | 1 | 10 | |
| (19) | 13 | 18 | Blue cell | 1.01 | 60.84/121.68 | 30.42 | 1 | 10 | 360 |
| | | | Red cell | 1.01 | 60.84/121.68 | 30.42 | 1 | 2 | |
| | | | Geen cell | 1.01 | 60.84/121.68 | 30.42 | 1 | 20 | |
| (20) | 13 | 18 | Blue cell | 1.01 | 60.84/121.68 | 30.42 | 1 | 20 | 480 |

*Table 3 continued on next page*

*Table 3 continued*

| Set# | $w$ | $h$ | Cell type | $A_0/\bar{A}$ | $J_a$ , $J_b$ | $J_l$ | $\lambda$ | $E_s$ | $T$ |
|---|---|---|---|---|---|---|---|---|---|
| | | | Red cell | 1.04 | 243.36 /486.72 | 121.68 | 1 | 0.25 | |
| | | | Green cell | 1.04 | 243.36 /486.72 | 121.68 | 1 | 2.5 | |
| (21) | 13 | 18 | Blue cell | 1.04 | 243.36 /486.72 | 121.68 | 1 | 2.5 | 480 |
| | | | Red cell | 1.04 | 243.36 /486.72 | 121.68 | 1 | 0.5 | |
| | | | Green cell | 1.04 | 243.36 /486.72 | 121.68 | 1 | 5.0 | |
| (22) | 13 | 18 | Blue cell | 1.04 | 243.36 /486.72 | 121.68 | 1 | 5.0 | 480 |
| | | | Red cell | 1.04 | 243.36 /486.72 | 121.68 | 1 | 1 | |
| | | | Green cell | 1.04 | 243.36 /486.72 | 121.68 | 1 | 10 | |
| (23) | 13 | 18 | Blue cell | 1.04 | 243.36 /486.72 | 121.68 | 1 | 10 | 480 |
| | | | Red cell | 1.04 | 243.36 /486.72 | 121.68 | 1 | 2 | |
| | | | Green cell | 1.04 | 243.36 /486.72 | 121.68 | 1 | 20 | |
| (24) | 13 | 18 | Blue cell | 1.04 | 243.36 /486.72 | 121.68 | 1 | 20 | 600 |

in *Figure 2a*. The set# (10) corresponds to the simulation with surface elasticity (*Figure 2b*). The set # (11) and (12) correspond to the simulations of supracellular myosin cable (*Figure 2c*). Note that the surface elasticity was evaluated by *Equation 2*.

*Table 3* shows the parameters for the simulations shown in *Figures 4 and 5*.

Parameter labels represent the same ones with *Table 1*. The set# (13), (17), and (21) correspond with $E_s = 2.5$, (14), (18), and (22) correspond with $E_s = 5.0$, (15), (19), and (23) correspond with $E_s = 10$, and (16), (20), and (24) correspond with $E_s = 20$ in the figures. Parameters $J_a$ and $J_b$ values for $J_b/J_l = 2$ and $J_b/J_l = 4$ were displayed separated by a slash. Note that the surface elasticity was evaluated by *Equation 3*, and $P_{a0}$ was set same with the initial $P_a$ value for the red and green cells and 0 for the blue cells.

*Table 4* shows the parameters for the simulations shown in *Figure 4—figure supplement 1*.

Parameter labels represent the same ones with *Table 1*. The set# (25), (27), and (29) correspond with $J_a/J_b = 1.4$, and (26), (28), and (30) correspond with $J_a/J_b = 1.6$ in *Figure 4—figure supplement 1*.

*Table 5* shows the parameters for the simulations shown in *Figure 5—figure supplement 1*.

Parameter labels represent the same ones with *Table 1*. The variable $x$ varied from 2.5 to 20, and $X$ was set according to $A_0/\bar{A}$ and $E_s$ in the same way with *Table 3*.

*Table 6* shows the parameters for the simulations shown in *Figure 5—figure supplement 2*.

In *Table 6*, $J_a$ indicates the contact energy between the cytosol and the outer body fluid and $J_{aECM}$ indicates the contact energy between cytosol and the apical ECM. Variable $x_1$ varied from –1 to 0.5, $x_2$ varied from 2.5 to 10, and $X$ was set according to $A_0/\bar{A}$ and $E_s$.

## Energy landscape with respect to cell shape

To analyze the energy with respect to cell shape, we prepared a simple 2D model of single cell represented by four vertices and four edges. Apical and basal edges were straight, while a lateral edge was straight but another lateral edge was assigned a constant curvature. Let $v_1$, $v_2$, $v_3$, and $v_4$

**Table 4.** Parameters for gradient apical contractility.

| Set# | $w$ | $h$ | Cell type | $A_0/\bar{A}$ | $J_a$ | $J_l$ | $J_b$ | $\lambda$ | $E_s$ | $T$ |
|---|---|---|---|---|---|---|---|---|---|---|
| | | | Red cell | 1.0025 | 15.21 /30.42 | 7.605 | 15.21 /30.42 | 1 | 0 | |
| | | | Green cell | 1.0025 | 15.21 /30.42 | 7.605 | 15.21 /30.42 | 1 | 0 | |
| | | | Paler blue cell | 1.0025 | 21.294 /42.588 | 7.605 | 15.21 /30.42 | 1 | 0 | |
| | | | Middle blue cell | 1.0025 | 24.336 /48.672 | 7.605 | 15.21 /30.42 | 1 | 0 | |
| (25) | 13 | 18 | Darker blue cell | 1.0025 | 25.857 /51.714 | 7.605 | 15.21 /30.42 | 1 | 0 | 30 |
| | | | Red cell | 1.0025 | 15.21 /30.42 | 7.605 | 15.21 /30.42 | 1 | 0 | |
| | | | Green cell | 1.0025 | 15.21 /30.42 | 7.605 | 15.21 /30.42 | 1 | 0 | |
| | | | Paler blue cell | 1.0025 | 24.336 /48.672 | 7.605 | 15.21 /30.42 | 1 | 0 | |
| | | | Middle blue cell | 1.0025 | 28.899 /57.798 | 7.605 | 15.21 /30.42 | 1 | 0 | |
| (26) | 13 | 18 | Darker blue cell | 1.0025 | 31.1805 /62.361 | 7.605 | 15.21 /30.42 | 1 | 0 | 30 |
| | | | Red cell | 1.01 | 60.084 /121.68 | 30.42 | 60.84 /121.68 | 1 | 0 | |
| | | | Green cell | 1.01 | 60.84 /121.68 | 30.42 | 60.84 /121.68 | 1 | 0 | |
| | | | Paler blue cell | 1.01 | 85.176 /170.354 | 30.42 | 60.84 /121.68 | 1 | 0 | |
| | | | Middle blue cell | 1.01 | 97.344 /194.688 | 30.42 | 60.84 /121.68 | 1 | 0 | |
| (27) | 13 | 18 | Darker blue cell | 1.01 | 103.428 /206.856 | 30.42 | 60.84 /121.68 | 1 | 0 | 60 |
| | | | Red cell | 1.01 | 60.84 /121.68 | 30.42 | 60.84 /121.68 | 1 | 0 | |
| | | | Green cell | 1.01 | 60.84 /121.68 | 30.42 | 60.84 /121.68 | 1 | 0 | |
| | | | Paler blue cell | 1.01 | 97.344 /194.688 | 30.42 | 60.84 /121.68 | 1 | 0 | |
| | | | Middle blue cell | 1.01 | 115.596 /231,192 | 30.42 | 60.84 /121.68 | 1 | 0 | |
| (28) | 13 | 18 | Darker blue cell | 1.01 | 124.722 /249.444 | 30.42 | 60.84 /121.68 | 1 | 0 | 60 |

*Table 4 continued on next page*

*Table 4 continued*

| Set# | w | h | Cell type | $A_0/\bar{A}$ | $J_a$ | $J_l$ | $J_b$ | $\lambda$ | $E_s$ | $T$ |
|---|---|---|---|---|---|---|---|---|---|---|
| | | | Red cell | 1.04 | 243.36 /486.72 | 121.68 | 243.36 /486.72 | 1 | 0 | |
| | | | Green cell | 1.04 | 243.36 /486.72 | 121.68 | 243.36 /486.72 | 1 | 0 | |
| | | | Paler blue cell | 1.04 | 340.704 /681.408 | 121.68 | 243.36 /486.72 | 1 | 0 | |
| | | | Middle blue cell | 1.04 | 389.376 /778.752 | 121.68 | 243.36 /486.72 | 1 | 0 | |
| (29) | 13 | 18 | Darker blue cell | 1.04 | 413.712 /827.424 | 121.68 | 243.36 /486.72 | 1 | 0 | 120 |
| | | | Red cell | 1.04 | 243.36 /486.72 | 121.68 | 243.36 /486.72 | 1 | 0 | |
| | | | Green cell | 1.04 | 243.36 /486.72 | 121.68 | 243.36 /486.72 | 1 | 0 | |
| | | | Paler blue cell | 1.04 | 389.376 /778.752 | 121.68 | 243.36 /486.72 | 1 | 0 | |
| | | | Middle blue cell | 1.04 | 462.384 /924.768 | 121.68 | 243.36 /486.72 | 1 | 0 | |
| (30) | 13 | 18 | Darker blue cell | 1.04 | 498.888 /997.776 | 121.68 | 243.36 /486.72 | 1 | 0 | 120 |

denote the vertices, where an edge between $v_1$ and $v_4$ is the apical, an edge between $v_2$ and $v_3$ is the basal, an edge between $v_1$ and $v_2$ is the curved lateral, and an edge between $v_3$ and $v_4$ is the straight lateral edge. For a radius $r$ of the curvature, an angle $\theta$ can be determined as $2r\sin\theta = \|v_2 - v_1\|$ and so a length of the curved edge is $2r\theta$. Also, an area between a chord and an arc for $v_1$ and $v_2$ is $r^2\theta - 1/2\|v_2 - v_1\|r\cos\theta$, and an area $A$ in the cell is

$$A = r^2\theta - \frac{1}{2}\|v_2 - v_1\|r\cos\theta + \frac{1}{2}(v_1 \times v_2 + v_2 \times v_3 + v_3 \times v_4 + v_4 \times v_1). \tag{11}$$

In the same way with the cellular Potts model and the vertex model, the energy $\mathcal{H}$ of the cell was defined with the surface contractility and the area constraint,

$$\mathcal{H} = J_a\|v_1 - v_4\| + J_l(2r\theta + \|v_4 - v_3\|) + J_b\|v_3 - v_2\| + \lambda(A - A_0)^2, \tag{12}$$

where $J_a$, $J_l$, and $J_b$ denote the apical, lateral, and basal surface contractilities, $\lambda$ denotes the bulk modulus, and $A_0$ is the reference value. For an apical width, $v_1$ and $v_4$ positions were set, and for the lateral curvature, $v_2$ and $v_3$ positions were searched for a minimum energy. For the cell to be stable in a rectangular shape with the apical width $w = 13$, lateral height $h = 18$, 0 curvature, $\lambda = 1$, and the area reference value $A_0 = 1.02wh$, the surface contractilities were set as $J_b = \lambda(A_0 - wh)h$ and $J_l = \lambda(A_0 - wh)w$. The apical surface contractility $J_a$ was set $J_a = 1.6J_b$. The apical width and the curvature were varied from $w$ to $w/4$ and 0 to $0.5/h$ in 100 bins, respectively, and the energy landscape was plotted.

A gradient map was obtained from the energy landscape by Prewitt filter, and a path from the apical width $w$ and 0 curvature following the gradient was calculated numerically.

The cell shape searching and the path tracking were implemented by MATLAB custom scripts which are available at GitHub (https://doi.org/10.5281/zenodo.8354144; *Yamashita, 2023b*).

## Image acquisition

*Drosophila* eggs were collected at 25°C, and dechorionated embryos were mounted on a glass-bottom dish with glue and submerged under water. The lateral epidermis were imaged with a confocal

**Table 5.** Parameters for various cell heights.

| Set# | $w$ | $h$ | Cell type | $A_0/\bar{A}$ | $J_a$, $J_b$ | $J_l$ | $\lambda$ | $E_s$ | $T$ |
|---|---|---|---|---|---|---|---|---|---|
| | | | Red cell | 1.0025 | 10.14 /20.28 | 5.07 | 1 | $0.1 \times x$ | |
| | | | Green cell | 1.0025 | 10.14 /20.28 | 5.07 | 1 | $x$ | |
| (31) | 13 | 12 | Blue cell | 1.0025 | 10.14 /20.28 | 5.07 | 1 | $x$ | $X$ |
| | | | Red cell | 1.0025 | 20.28 /40.56 | 10.14 | 1 | $0.1 \times x$ | |
| | | | Green cell | 1.0025 | 20.28 /40.56 | 10.14 | 1 | $x$ | |
| (32) | 13 | 24 | Blue cell | 1.0025 | 20.28 /40.56 | 10.14 | 1 | $x$ | $X$ |
| | | | Red cell | 1.01 | 40.56 /81.12 | 20.28 | 1 | $0.1 \times x$ | |
| | | | Green cell | 1.01 | 40.56 /81.12 | 20.28 | 1 | $x$ | |
| (33) | 13 | 12 | Blue cell | 1.01 | 40.56 /81.12 | 20.28 | 1 | $x$ | $X$ |
| | | | Red cell | 1.01 | 81.12 /162.24 | 40.56 | 1 | $0.1 \times x$ | |
| | | | Green cell | 1.01 | 81.12 /162.24 | 40.56 | 1 | $x$ | |
| (34) | 13 | 24 | Blue cell | 1.01 | 81.12 /162.24 | 40.56 | 1 | $x$ | $X$ |
| | | | Red cell | 1.04 | 162.24 /324.48 | 81.12 | 1 | $0.1 \times x$ | |
| | | | Geen cell | 1.04 | 162.24 /324.48 | 81.12 | 1 | $x$ | |
| (35) | 13 | 12 | Blue cell | 1.04 | 162.24 /324.48 | 81.12 | 1 | $x$ | $X$ |
| | | | Red cell | 1.04 | 324.48 /648.96 | 162.24 | 1 | $0.1 \times x$ | |
| | | | Green cell | 1.04 | 324.48 /648.96 | 162.24 | 1 | $x$ | |
| (36) | 13 | 24 | Blue cell | 1.04 | 324.48 /648.96 | 162.24 | 1 | $x$ | $X$ |

microscope (Olympus FV1000) equipped with a ×60 oil immersion objective lens (PLAPON 60X0) at 25°. Z stack of 0.7 μm interval with slices of 0.125 μm x-y resolution was taken every 3 minutes.

## Cell tracking

The cells were manually tracked with Fiji plugin TrackMate manual tracking.

## Conversion of adherens junction into graph

For the geometrical tension inference, the cells were segmented manually and converted to the vertices and edges representation. The 3D image of cadherin was projected onto the x-y plane, and the adherens junction was traced manually. The trace was narrowed to 1 pixel boundary by Fiji plugin MorphoLibJ Segmentation Interactive Marker-controlled Watershed. In the segmented image, the boundary pixels were scanned and the vertices were allocated for pixels incident to more than two cells, that is, the boundary pixel was next to pixels in the cells. Next, all pairs of the cells were enumerated and checked if there were two vertices incident to both of the cells, and edges were allocated to

**Table 6.** Parameters for deformation with apical elasticity and cell-extracellular matrix adhesion.

| Set# | $w$ | $h$ | Cell type | $A_0/\bar{A}$ | $J_a$, $J_b$ | $J_{aECM}$ | $J_l$ | $\lambda$ | $E_s$ | $T$ |
|---|---|---|---|---|---|---|---|---|---|---|
| | | | Red cell | 1.0025 | 15.21/30.42 | 15.21/30.42 | 7.605 | 1 | $0.1x_2$ | |
| | | | Green cell | 1.0025 | 15.21/30.42 | $15.21x_1/30.42x_1$ | 7.605 | 1 | $x_2$ | |
| (36) | 13 | 18 | Blue cell | 1.0025 | 15.21/30.42 | 15.21/30.42 | 7.605 | 1 | $x_2$ | $X$ |
| | | | Red cell | 1.01 | 60.84/121.68 | 60.84/121.68 | 30.42 | 1 | $0.1x_2$ | |
| | | | Green cell | 1.01 | 60.84/121.68 | $60.84x_1/121.68x_1$ | 30.42 | 1 | $x_2$ | |
| (37) | 13 | 18 | Blue cell | 1.01 | 60.84/121.68 | 60.84/121.68 | 30.42 | 1 | $x_2$ | $X$ |
| | | | Red cell | 1.04 | 243.36/486.72 | 243.36/486.72 | 121.68 | 1 | $0.1x_2$ | |
| | | | Green cell | 1.04 | 243.36/486.72 | $243.36x_1/486.72x_1$ | 121.68 | 1 | $x_2$ | |
| (38) | 13 | 18 | Blue cell | 1.04 | 243.36/486.72 | 243.36/486.72 | 121.68 | 1 | $x_2$ | $X$ |

link those vertices. Pixels incident to both of the cells were assigned to the edge, and a depth of the brightest pixel in the stack for x and y of each pixel was read. When the depth differed by more than 3 from an average depth of adjacent edge pixels, the depth was masked as a noise. For each vertex, a plane was fitted by the least square method to x, y, and depths of the incident edges pixels, and depth of the plane at the vertex x and y position was assigned to the vertex. Vertices closer than 2 to each other were merged into a vertex.

## 3D Bayesian tension inference

The balance equation was obtained in a tangent plane for each vertex. For a vertex $v_0$, its adjacent vertices $v_1$, $v_2$, were collected and a plane was fitted to them by the least-square method. The vertices $v_0$, $v_1$, and edges between them $e_1$, $e_2$, were projected on the plane, and the projected edge $e'_1$, $e'_2$, were measured their lengths and angles. An axis was set parallel with $e'_1$, and another axis was also set orthogonal to the first axis. The balance equation was obtained with respect to the axes. A total force $f_1$ to move $v_0$ along the first axis was represented by an equation

$$f_1 = \sum_{i=1}^{k} -\cos(\theta_i)T_i + \frac{1}{2}\sum_{j=1}^{k}(|e'_{j+1}|\sin\theta_{j+1} - |e'_j|\sin\theta_j)P_j, \tag{13}$$

where $k$ is a number of the adjacent vertices, $\theta_i$ denotes an angle between the edge $e'_1$ (and thus the first axis) and the edge $e'_i$, $|e'_i|$ represents the length of $e'_i$, $e'_{k+1} = e'_1$, $T_i$ denotes a tension exerted on $e'_i$, and $P_j$ denotes a pressure in a cell containing edges $e'_j$ and $e'_{j+1}$. Note that the symbols $T_i$ and $P_j$ follows the *Ishihara and Sugimura, 2012* original work and should not be confused with the symbols which denoted the perimeter length and temperature. In the same way, a total force $f_2$ to move $v_0$ along the second axis was represented by an equation

$$f_2 = \sum_{i=1}^{k} -\sin(\theta_i)T_i - \frac{1}{2}\sum_{j=1}^{k}(|e'_{j+1}|\cos\theta_{j+1} - |e'_j|\cos\theta_j)P_j. \tag{14}$$

The coefficients for $T_i$ is and $P_j$ s were combined in an $n \times m$ matrix **A** so that **Ap** represents the forces, where $n$ is a number of the equations, $m$ is a number of the tensions and pressures, and **p** is an $m$-dimensional vector composed of $T_i$ and $P_j$.

The equations above are based on an assumption that an adherens junction is responsible for the edge tension and it makes an arc so that its angle of incidence to the vertex is parallel with the plane tangent at the vertex. We also approximated the tangent plane with the plane fitted to the adjacent vertices.

The Bayesian tension inference is to find the most likely distribution of tensions and pressures which makes **Ap** reasonably small. Since *Equations 13 and 14* only evaluate the balance among the forces, it cannot estimate an absolute value but a relative value of the tension and pressure. It was numerically estimated as below. Let a parameter $\mu = \sigma^2/\omega^2$, where $\sigma$ represents errors of the equations and $\omega$ is a variance of the tensions and pressures distribution. The parameter μ was searched for a minimization of Akaike Bayesian information criterion (ABIC). To calculate ABIC, we prepared $\tau = \mu^{1/2}$ and

$$\mathbf{S} = \begin{pmatrix} \mathbf{A} & \mathbf{0} \\ \tau\mathbf{B} & \tau\mathbf{g} \end{pmatrix}, \tag{15}$$

where

$$\mathbf{g} = \begin{pmatrix} 1 \\ \vdots \\ 1 \\ 0 \\ \vdots \\ 0 \end{pmatrix} \begin{array}{l} \left.\begin{array}{c} \\ \\ \\ \end{array}\right\} \text{number of edges with tension, } n_e \\ \left.\begin{array}{c} \\ \\ \\ \end{array}\right\} \text{number of cells with pressure, } n_c \end{array}, \tag{16}$$

and **B** was a diagonal matrix of **g**. Then **S** was decomposed by a QR factorization, **S = QR**. Let **R** be expressed as

$$\mathbf{R} = \begin{pmatrix} \mathbf{H} & \mathbf{h} \\ \mathbf{0} & h \end{pmatrix}, \tag{17}$$

and $\mathbf{d}_{H1}$, $\mathbf{d}_{H2}$, be an array of absolute value of non-zero diagonal elements of **H**. The ABIC was calculated by

$$\text{ABIC} = n_{km} + n_{km}\log\frac{2\pi h^2}{n_{km}} + 2\sum_i \log\mathbf{d}_{Hi} - n_e\log\mu + 2n_p, \tag{18}$$

where $n_{km} = n - n_c + 1$ and $n_p$ is a number of parameters, here $n_p = 2$. After the search of μ which minimizes ABIC, **p** was evaluated by calculating

$$\mathbf{p} = \mathbf{H}^{-1}\mathbf{h}, \tag{19}$$

where $\mathbf{H}^{-1}$ is the Moore–Penrose pseudo inverse matrix of **H**.

For the basis of the calculation, see *Ishihara and Sugimura, 2012*.

## Cell apical area calculation

For a cell, a plane was fitted to incident vertices of the cell. The vertices and edges between them were projected on the plane, and an area of the polygon was calculated as the cell apical surface area.

## Implementation of the tissue apical surface analysis

The conversion from the segmented image to the vertices and edges representation, the geometrical tension inference, and the apical area calculation was implemented by MATLAB custom scripts which are available at GitHub (https://doi.org/10.5281/zenodo.8320862; *Yamashita, 2023c*).

## Acknowledgements

We thank Shigeo Hayashi and members of Laboratory for Morphogenetic Signaling for providing the microscopic data and discussion. This work was supported by RIKEN Special Postdoctoral Researchers Program (SY).

## Additional information

### Funding

| Funder | Grant reference number | Author |
|---|---|---|
| Japan Society for the Promotion of Science | JSPS Overseas Research Fellowship, 201860572 | Satoshi Yamashita |
| RIKEN Center for Biosystems Dynamics Research | RIKEN SPDR Program, K2031052 | Satoshi Yamashita |

The funders had no role in study design, data collection and interpretation, or the decision to submit the work for publication.

### Author contributions

Satoshi Yamashita, Conceptualization, Resources, Software, Funding acquisition, Visualization, Writing - original draft, Writing – review and editing; Shuji Ishihara, Formal analysis, Methodology, Writing – review and editing; François Graner, Formal analysis, Supervision, Methodology, Project administration, Writing – review and editing

### Author ORCIDs

Satoshi Yamashita ⓘ https://orcid.org/0000-0002-1271-4911

Joint Public Review: https://doi.org/10.7554/eLife.93496.4.sa1
Author response https://doi.org/10.7554/eLife.93496.4.sa2

## Additional files

### Supplementary files
MDAR checklist

### Data availability

Codes developed and used in this study were deposited to GitHub as mentioned in the Methods.

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
