## [Editor Report · eLife Assessment]

The results from this study, which investigates the mechanisms necessary for initiating tissue invagination using a cellular Potts modelling approach, suggest that apical constriction is not sufficient to drive the process by itself. The study highlights how choices inherent to modeling – such as permitting straight or curved cell edges – may affect the outcome of simulations and, consequently, their biophysical interpretation. Despite **incomplete** evidence supporting their major claims due to a rather coarse-grained exploration of the model, this work is **useful** for biophysicists investigating complex tissue deformation through computational frameworks.

---

## [Referee Report · Joint Public Review]

Satoshi Yamashita et al., investigate the physical mechanisms driving tissue bending using the cellular Potts Model, starting from a planar cellular monolayer. They argue that apical length-independent tension control alone cannot explain bending phenomena in the cellular Potts Model, contrasting with previous works, particularly Vertex Models. They conclude that an apical elastic term, with zero rest value (due to endocytosis/exocytosis), is necessary to achieve apical constriction and that tissue bending can be enhanced by adding a supracellular myosin cable. Additionally, a very high apical elastic constant promotes planar tissue configurations, opposing bending.

Strengths:

- The finding of the required mechanisms for tissue bending in the cellular Potts Model provides a natural alternative for studying bending processes in situations with highly curved cells.

- Despite viewing cellular delamination as an undesired outcome in this particular manuscript, the model's capability to naturally allow T1 events might prove useful for studying cell mechanics during out-of-plane extrusion.

[Editors' note: The previous reviews have not been updated, as the changes to the manuscript were restricted to refining the text. The authors addressed all of the minor points raised by the reviewers. Some of the major points such as the lack of a summary quantification still stand. The previous reviews are here: https://doi.org/10.7554/eLife.93496.2.sa1]

---

## [Author Response]

The following is the authors’ response to the previous reviews.

**Public Reviews:**

**Reviewer #1 (Public Review):**
Summary:Satoshi Yamashita et al., investigate the physical mechanisms driving tissue bending using the cellular Potts Model, starting from a planar cellular monolayer. They argue that apical length-independent tension control alone cannot explain bending phenomena in the cellular Potts Model, contrasting with previous works, particularly Vertex Models. They conclude that an apical elastic term, with zero rest value (due to endocytosis/exocytosis), is necessary to achieve apical constriction, and that tissue bending can be enhanced by adding a supracellular myosin cable. Additionally, a very high apical elastic constant promotes planar tissue configurations, opposing bending.Strengths:- The finding of the required mechanisms for tissue bending in the cellular Potts Model provides a natural alternative for studying bending processes in situations with highly curved cells.- Despite viewing cellular delamination as an undesired outcome in this particular manuscript, the model's capability to naturally allow T1 events might prove useful for studying cell mechanics during out-of-plane extrusion.

We thank the reviewer for the careful comments and suggestions.

Weaknesses:- The authors claim that the cellular Potts Model (CPM) is unable to achieve the results of the vertex model (VM) simulations due to naturally non-straight cellular junctions in the CPM versus the VM. The lack of a substantial comparison undermines this assertion. None of the references mentioned in the manuscript are from a work using vertex model with straight cellular junctions, simulating apical constriction purely by a enhancing a length-independent apical tension. Sherrard et al and Pérez-González et al. use 2D and 3D Vertex Models, respectively, with a "contractility" force driving apical constriction. However, their models allow cell curvature. Both references suggest that the cell side flexibility of the CPM shouldn't be the main issue of the "contractility model" for apical constriction.

We appreciate the comment.

For the reports by Sherrard et al and Pérez-Gonález et al, lack of the cell rearrangement (T1 transition) might have caused the difference. Other than these, Muñoz et al. (doi:10.1016/j.jbiomech.2006.05.006), Polyakov et al. (doi:10.1016/j.bpj.2014.07.013), Inoue et al.

(doi:10.1007/s10237-016-0794-1), Sui et al.

(doi:10.1038/s41467-018-06497-3), and Guo et al. (doi:10.7554/eLife.69082) used simulation models with the straight lateral surface.

We updated an explanation about the difference between the vertex model and the cellular Potts model in the discussion.

P12L318 “An edge in the vertex model can be bent by interpolating vertices or can be represented with an arc of circle (Brakke, 1992). Even in cases where vertex models were extended to allow bent lateral surfaces, the model still limited cell rearrangement and neighbor changes (Pérez-González et al., 2021), limiting the cell delamination. Thus the difference in simulation results between the models could be due to whether the cell rearrangement was included or not. However, it is not clear how the absence of the cell rearrangement affected cell behaviors in the simulation, and it shall be studied in future. In contrast to the vertex model, the cellular Potts model included the curved cell surface and the cell rearrangement innately, it elucidated the importance of those factors.”

- The myosin cable is assumed to encircle the invaginated cells. Therefore, it is not clear why the force acts over the entire system (even when decreasing towards the center), and not locally in the contour of the group of cells under constriction. The specific form of the associated potential is missing. It is unclear how dependent the results of the manuscript are on these not-well-motivated and model-specific rules for the myosin cable.

A circle radius decreases when the circle perimeter shrinks, and this was simulated with the myosin cable moving toward the midline in the cross section.

We added an explanation in the introduction and the results.

P2L74 “In the same way with the contracting circumferential myosin belt in a cell decreasing the cell apical surface, the circular supracellular myosin cable contraction decreases the perimeter, the radius of the circle, and an area inside the circle.”

P6L197 “In the cross section, the shrinkage of the circular supracellular myosin cable was simulated with a move of adherens junction under the myosin cable toward the midline.”

- The authors are using different names than the conventional ones for the energy terms. Their current attempt to clarify what is usually done in other works might lead to further confusion.

The reviewer is correct. However we named the energy terms differently because the conventional naming would be misleading in our simulation model.

We added an explanation in the results.

P4L140 “Note that the naming for the energy terms differs from preceding studies. For example, Farhadifar et al. (2007) named a surface energy term expressed by a proportional function "line tensions" and a term expressed by a quadratic function "contractility of the cell perimeter". In this study, however, calling the quadratic term "contractility" would be misleading since it prevents the contraction when < _0. Therefore we renamed the terms accordingly.”

**Reviewer #2 (Public Review):**
Summary:In their work, the Authors study local mechanics in an invaginating epithelial tissue. The work, which is mostly computational, relies on the Cellular Potts model. The main result shows that an increased apical "contractility" is not sufficient to properly drive apical constriction and subsequent tissue invagination. The Authors propose an alternative model, where they consider an alternative driver, namely the "apical surface elasticity".Strengths:It is surprising that despite the fact that apical constriction and tissue invagination are probably most studied processes in tissue morphogenesis, the underlying physical mechanisms are still not entirely understood. This work supports this notion by showing that simply increasing apical tension is perhaps not sufficient to locally constrict and invaginate a tissue.

We thank the reviewer for the careful comments.

Weaknesses:Although the Authors have improved and clarified certain aspects of their results as suggested by the Reviewers, the presentation still mostly relies on showing simulation snapshots. Snapshots can be useful, but when there are too many, the results are hard to read. The manuscript would benefit from more quantitative plots like phase diagrams etc.

We agree with the comment.

However, we could not make the qualitative measurement for the phase diagram since (1) the measurement must be applicable to all simulation results, and (2) measured values must match with the interpretation of the results. To do so, the measurement must distinguish a bent tissue, delaminated cells, a tissue with curved basal surface and flat apical surface, and a tissue with closed invagination. Such measurement is hardly designed.

**Recommendations for the authors:**

**Reviewing Editor (Recommendations For The Authors):**
I see that the authors have worked on improving their paper in the revision. However, I agree with both reviewer #1 and reviewer #2 that the presentation and discussion of findings could be clearer.Concrete recommendations for improvement:(1) I find the observation by reviewer #1 on cell rearrangement very illuminating: It is indeed another key difference between the Cellular Potts Model that the authors use compared to typical Vertex Models, and could very well explain the different model outcomes. The authors could expand on the discussion of this point.

We updated an explanation about the difference between the vertex model and the cellular Potts model in the discussion.

P12L318 “An edge in the vertex model can be bent by interpolating vertices or can be represented with an arc of circle (Brakke, 1992). Even in cases where vertex models were extended to allow bent lateral surfaces, the model still limited cell rearrangement and neighbor changes (Pérez-González et al., 2021), limiting the cell delamination. Thus the difference in simulation results between the models could be due to whether the cell rearrangement was included or not. However, it is not clear how the absence of the cell rearrangement affected cell behaviors in the simulation, and it shall be studied in future. In contrast to the vertex model, the cellular Potts model included the curved cell surface and the cell rearrangement innately, it elucidated the importance of those factors.”

(2) In lines 161-164, the authors write "Some preceding studies assumed that the apical myosin generated the contractile force (Sherrard et al, 2010: Conte et al., 2012; Perez-Mockus et al., 2017; Perez-Gonzalez et al., 2021), while others assumed the elastic force (Polyakov et al., 2014; Inoue et al. 2016; Nematbakhsh et al., 2020)."Similarly, in lines 316-319 the authors write "In the preceding studies, the apically localized myosin was assumed to generate either the contractile force (Sherrard et al, 2010: Conte et al., 2012; Perez-Mockus et al., 2017; Perez-Gonzalez et al., 2021), or the elastic force (Polyakov et al., 2014; Inoue et al. 2016; Nematbakhsh et al., 2020)."The phrasing here is poor, as it suggests that the latter three studies (Polyakov et al., 2014; Inoue et al. 2016; Nematbakhsh et al., 2020) do not use the assumption that apical myosin generated contractile forces. This is wrong. All three of these studies do in fact assume apical surface contractility mediated by myosin. In addition, they also include other factors such as elastic restoring forces from the cell membrane (but not mediated by myosin as far as I understand).These statements should be corrected.

We named the energy term expressed with the proportional function “contractility” and the energy term expressed with the quadratic function “elasticity”. Here we did not define what biological molecules correspond with the contractility or the elasticity.

For the three studies, the effect of myosin was expressed by the quadratic function, and Polyakov et al. (2014) named it “springlike elastic properties”, Inoue et al. (2016) named it “Apical circumference elasticity”, and Nematbakhsh et al. (2020) named it “Actomyosin contractility”. To explain that the for generated by myosin was expressed with the quadratic function in these studies, we wrote that they “assumed the elastic force”.

We assumed the myosin activity to be approximated with the proportional function in later parts and proposed that the membrane might be expressed with the quadratic function and responsible for the apical constriction based on other studies.

To clarify this, we added it to the results.

P4L175 “Some preceding studies assumed that the apical myosin generated the contractile force (Sherrard et al., 2010; Conte et al., 2012; Perez-Mockus et al., 2017; Pérez-González et al., 2021), while the others assumed the myosin to generate the elastic force (Polyakov et al., 2014; Inoue et al., 2016; Nematbakhsh et al., 2020).”

(3) Lines 294-296: The phrasing suggests that the "alternative driving mechanism" consists of apical surface elasticity remodelling alone. This is not true, it's an additional mechanism, not an alternative. The authors' model works by the combined action of increased apical surface contractility and apical surface elasticity remodelling (and the effect can be strengthened by including a supracellular actomyosin cable).

We agree with the comment that the surface remodeling is not solely driving the apical constriction but with myosin activity. However, if we wrote it as an additional mechanism, it might look like that both the myosin activity alone and the surface remodeling alone could drive the apical constriction, and they would drive it better when combined together. So we replaced “mechanism” with “model”.

P12L311 “In this study, we demonstrated that the increased apical surface contractility could not drive the apical constriction, and proposed the alternative driving model with the apical surface elasticity remodeling.”

(4) In general, the part of the results section encompassing equations 1-5 should more explicitly state which equations were used in all simulations (Eqs1+5), and which ones were used only for certain conditions (Eqs2+3+4).

We added it as follows.

P4L153 “While the terms Equation 1 and Equation 5 were included in all simulations since they were fundamental and designed in the original cellular Potts model (Graner and Glazier, 1992), the other terms Equation 2-Equation 4 were optional and employed only for certain conditions.”

(5) Lines 150-152: Please state which parameters were examined. I assume Equation 4 was also left out of this initial simulation, as it is the potential energy of the actomyosin cable that was only included in some simulations.

We added it as follows.

P4L163 “The term Equation 4 was not included either. For a cell, its compression was determined by a balance between the pressure and the surface tension, i.e., the heigher surface tension would compress the cell more. The bulk modulus 𝜆 was set 1, the lateral cell-cell junction contractility 𝐽_𝑙 was varied for different cell compressions, and the apical and basal surface contractilities 𝐽_𝑎 and 𝐽_𝑏 were varied proportional to 𝐽_𝑙.”

(6) Lines 118-122: The sentence is very long and hard to parse. I suggest the following rephrasing:“In this study, we assumed that the cell surface tension consisted of contractility and elasticity. We modelled the contractility as constant to decrease the surface, but not dependent on surface width or strain. We modelled the elasticity as proportional to the surface strain, working to return the surface to its original width."

We updated the explanation as follows.

P3L121 “In this study, we assumed that the cell surface tension consisted of contractility and elasticity. We modeled the contractility as a constant force to decrease the surface, but not dependent on surface width or strain. We modeled the elasticity as a force proportional to the surface strain, working to return the surface to its original width.”

(7) Lines 270-274: Another long sentence that is difficult to understand.Suggested rephrasing:"Note that the supracellular myosin cable alone could not reproduce the apical constriction (Figure 2c), and cell surface elasticity in isolation caused the tissue to stay almost flat. However, combining both the supracellular myosin cable and the cell surface elasticity was sufficient to bend the tissue when a high enough pulling force acted on the adherens junctions."

We updated the sentence as follows.

P9L287 “Note that the supracellular myosin cable alone could not reproduce the apical constriction (Figure 2c), and that with some parameters the modified cell surface elasticity kept the tissue almost flat (Figure 4). However, combining both the supracellular myosin cable and the cell surface elasticity made a sharp bending when the pulling force acting on the adherens junction was sufficiently high.”

(8) Lines 434-435: Unclear what is meant with sentence starting with "Rest of sites"

We update the sentence as follows.

P17L456 “At the initial configuration and during the simulation, sites adjacent to medium and not marked as apical are marked as basal.”

(9) Fixing typos and other minor grammar and wording changes would improve readability. Following is a list in order of appearance in the text with suggestions for improvement.

We greatly appreciate the careful editing, and corrected the manuscript accordingly.

Line 14: "a" is not needed in the phrase "increased a pressure"Line 15: "cell into not the wedge shape" --"cell not into the wedge shape" In fact it might be better to flip the sentence around to say, e.g. "making the cells adopt a drop shape instead of the expected wedge shape".Line 24: "cells decrease its apical surface" --"cells decrease their apical surface"Line 25: instead of "turn into wedge shape", a more natural-sounding expression could be "adopt a wedge shape"Line 28: "which crosslink and contract" --because the subject is the singular "motor protein", the verb tense needs to be changed to "crosslinks and contracts"Line 29: I suggest to use the definite article "the" before "actin filament network" as this is expected to be a known concept to the reader.Line 31: "adherens junction and tight junction" --use the plural, because there are many per cell: "adherens junctions and tight junctions"Line 42: "In vertebrate" --"In vertebrates"Line 46: "Since the interruption to" --"Since the interruption of"Line 56: "the surface tension of the invaginated cells were" --since the subject is "the surface tension", the verb "were" needs to be changed to "was" Line 63: "extra cellular matrix" --generally written as "extracellular matrix" without the first spaceLine 66: "many epithelial tissues" --"in many epithelial tissues"Line 70: "This supracellular cables" --"These supracellular cables"Line 72: "encircling salivary gland" --either "encircling the salivary gland" or "encircling salivary glands"Lines 76-77: "investigated a cell physical property required" --"investigated what cell physical properties were required"Line 78: "was another framework" --"is another framework" (it is a generally and currently valid true statement, so use the present tense)Line 79: "simulated an effect of the apically localized myosin" --for clarity, I suggest rephrasing as "simulated the effect of increased apical contractility mediated by apically localized myosin"Similarly, in Line 80: "did not reproduce the apical constriction" --"did not reproduce tissue invagination by apical constriction", as technically the cells in the model do reduce their apical area, but fail to invaginate as a tissue.Line 82: "we found that a force" --"we found that the force"Line 101: "apico-basaly" --"apico-basally"Lines 107-108: "in order to save a computational cost" --"in order to save on computational cost"Line 114: "Therefore an area of the cell" --"Therefore the interior area of the cell"Line 139: "formed along adherens junction" --"formed along adherens junctions"Line 166: "we ignored an effect" --"we ignored the effect"Line 167: "and discussed it later" --"and discuss it later"Lines 167-168: "an experiment with a cell cultured on a micro pattern showed that the myosin activity was well corresponded by the contractility" --"an experiment with cells cultured on a micro pattern showed that the myosin activity corresponded well to the contractility"Line 172: "success of failure" --"success or failure"Figure 1 caption: "none-polar" --"non-polarized"; "reg" --"red"Line 179: "To prevented the surface" --"To prevent the surface"Line 180: "It kept the cells surface" --"It kept the cells' surface" (apostrophe missing)Line 181: "cells were delaminated and resulted in similar shapes" --"cells were delaminated and adopted similar shapes"Line 190: "To investigate what made the difference" --"To investigate the origin of the difference"Line 203: For clarity, I would suggest to add more specific wording. "the pressure, and a difference in the pressure between the cells resulted in" --"the internal pressure due to cell volume conservation, and a difference in the pressure between the contracting and non-contracting cells resulted in"Line 206: "by analyzing the energy with respect to a cell shape" --"by analyzing the energy with respect to cell shape"Line 220: "indicating that cell could shrink" --"indicating that a cell could shrink"Line 224: For clarity, I would suggest more specific wording "lateral surface, while it seems not natural for the epithelial cells" --"lateral surface imposed on the vertex model, a restriction that seems not natural for epithelial cells"Line 244: "succeeded in invaginating" --"succeeding in invaginating"Line 247: "were checked whether the cells" --"were checked to assess whether the cells"Line 250: "cells became the wedge shape" --"cells adopted the wedge shape"Line 286: "there were no obvious change in a distribution pattern" --"there was no obvious change in the distribution pattern"Lines 296-297: "When the cells were assigned the high apical surface contractility, the cells were rounded" --"When the cells were assigned a high apical surface contractility, the cells became rounded"Line 298: "This simulation results" --"These simulation results"Lines 301-302: I suggest to increase clarity by somewhat rephrasing. "Even when the vertex model allowed the curved lateral surface, the model did not assume the cells to be rearranged and change neighbors" --"Even in cases where vertex models were extended to allow curved lateral surfaces, the model still limited cell rearrangement and neighbor changes"Line 326: "high surface tension tried to keep" --"high surface tension will keep"Line 334: "In many tissue" --"In many tissues"Line 345: "turned back to its original shape" --"turned back to their original shape" (subject is the plural "cells")Lines 348-349: "resembles the result of simulation" --"resembles the result of simulations"Line 352: "how the myosin" --"how do the myosin"Line 356: "it bears the surface tension when extended and its magnitude" What does the last "its" refer to? The surface tension?Line 365: "the endocytosis decrease" --"the endocytosis decreases"Line 371: "activatoin" --"activation"Line 374 "the cells undergoes" --"the cells undergo"Line 378: "entier" --"entire"Line 389: "individual tissue accomplish" --"individual tissues accomplish"Line 423: "is determined" --"are determined" (subject is the plural "labels")Line 430: "phyisical" --"physical"Table 6 caption: "cell-ECN" --cell-ECMLine 557: "do not confused" --"should not be confused"
**Reviewer #1 (Recommendations For The Authors):**
- The phrase "In addition, the encircling supracellular myosin cable largely promoted the invagination by the apical constriction, suggesting that too high apical surface tension may keep the epithelium apical surface flat." is not clear to me. It sounds contradictory.

This finding was unexpected and surprising for us too. However, it is actually not contradictory since stronger surface tension will make the surface flatter in general. Figure 4 shows the flat apical surface with the wedge shape cells for the too strong apical surface tension. On the other hand, the supracellular myosin cable promoted the cell shape changes without raising the surface tension, and thus it could make a sharp bending (Figure 5).

We updated the explanation for the effect of the supracellular myosin cable as follows.

P2L74 “In the same way as the contracting circumferential myosin belt in a cell decreasing the cell apical surface, the circular supracellular myosin cable contraction decreases the perimeter, the radius of the circle, and an area inside the circle.”

P6L197 “In the cross section, the shrinkage of the circular supracellular myosin cable was simulated with a move of adherens junction under the myosin cable toward the midline.”

- Even when the authors now avoid to say "in contrast to vertex model simulations" in pg.4, in the next section there is still the intention to compare VM to CPM. Idem in the Discussion section. The conclusion in that section is that the difference between the results arising with VM (achieving the constriction) and the CPM (not achieving the constriction, and leading to cell delamination) are due to the straight lateral surfaces. However, Sherrard et at could achieve the constriction with an enhanced apical surface contractility using a 2D VM that allows curvatures. Therefore, I don't think the main difference is given by the deformability of the lateral surfaces. Instead, it might be due to the facility of the CPM to drive cellular rearrangements, coupled to specific modeling rules such as the permanent lost of the "apical side" once a delamination occurs and the boundary conditions. A clear example is the observation of loss of cell-cell adherence when all the tensions are set the same. Instead, in a VM cells conserve their lateral neighbors in the uniform tension regime (Sherrard et at). Is it noteworthy that the two mentioned works using vertex models to achieve apical constriction (Sherrard et at. (2D) and Pérez-González (3D) et al.) seem to neglect T1 transitions. I specifically think the added discussion on the impact of the T1 events (fundamental for cell delamination) is quite poor. A more detailed description would help justify the differences between model outcomes.

We updated an explanation about the difference between the vertex model and the cellular Potts model in the discussion.

P12L318 “ An edge in the vertex model can be bent by interpolating vertices or can be represented with an arc of circle (Brakke, 1992). Even in cases where vertex models were extended to allow bent lateral surfaces, the model still limited cell rearrangement and neighbor changes (Pérez-González et al., 2021), limiting the cell delamination. Thus the difference in simulation results between the models could be due to whether the cell rearrangement was included or not. However, it is not clear how the absence of the cell rearrangement affected cell behaviors in the simulation, and it shall be studied in future. In contrast to the vertex model, the cellular Potts model included the curved cell surface and the cell rearrangement innately, it elucidated the importance of those factors.”

- Fig6c: cell boundary colors are quite difficult to see.

The images were drawn by custom scripts, and those scripts do not implement a method to draw wide lines.

- Title Table 1: "epitherila".

We corrected the typo.

**Reviewer #2 (Recommendations For The Authors):**
The Authors have addressed most of my initial comments. In my opinion, the results could be better represented. Overall, the manuscript contains too many snapshots that are hard to read. I am sure the Authors could come up with a parameter that would tell the overall shape of the tissue and distinguish between a proper invagination and delamination. Then they could plot this parameter in a phase diagram using color plots to show how varying values of model parameters affects the shape. Presentation aside, I believe the manuscript will be a valuable piece of work that will be very useful for the community of computational tissue mechanics.

We agree with the comment.

However, we could not make a suitable qualitative measurement method. For the phase diagrams, the measurement must be applicable to simulation results, otherwise each figure introduce a new measurement and a color representation would just redraw the snapshots but no comparison between the figures. So the different measurements would make the figures more difficult to read.

The single measurement must distinguish the cell delamination by the increased surface contractility from the invagination by the modified surface elasticity and the supracellular contractile ring, even though the center cells were covered by the surrounding cells and lost contact with apical side extracellular medium in both cases.

With the center of mass, the delaminated cells would return large values because they were moved basally. With the tissue basal surface curvature, it would not measure if the tissue apical surface was also curved or kept flat. If the phase diagram and interpretation of the simulation results do not match with each other, it would be misleading.

A measurement meeting all these conditions was hardly designed.